# FOUNDATION MODELS FOR CAUSAL INFERENCE VIA PRIOR-DATA FITTED NETWORKS

**Yuchen Ma**[*][†]
LMU Munich & MCML
yuchen.ma@lmu.de

**Dennis Frauen**[*]
LMU Munich & MCML
frauen@lmu.de

**Emil Javurek**
LMU Munich & MCML
emil.javurek@lmu.de

**Stefan Feuerriegel**
LMU Munich & MCML
feuerriegel@lmu.de

## ABSTRACT

Prior-data fitted networks (PFNs) have recently been proposed as a promising way to train tabular foundation models. PFNs are transformers that are pre-trained on synthetic data generated from a prespecified prior distribution and that enable Bayesian inference through in-context learning. In this paper, we introduce *CausalFM*, a comprehensive framework for training PFN-based foundation models in various causal inference settings. First, we formalize the construction of Bayesian priors for causal inference based on structural causal models (SCMs) in a principled way and derive necessary criteria for the validity of such priors. Building on this, we propose a novel family of prior distributions using causality-inspired Bayesian neural networks that enable CausalFM to perform Bayesian causal inference in various settings, including for back-door, front-door, and instrumental variable adjustment. Finally, we instantiate CausalFM and explicitly train models to perform in-context learning in these settings. We show that CausalFM achieves competitive in-context learning performance even when compared to baselines that are specifically trained for the task at hand. In sum, our framework can be used as a general recipe to train foundation models for various causal inference settings. In contrast to the current state-of-the-art in causal inference, CausalFM offers a novel paradigm with the potential to fundamentally change how practitioners perform causal inference in medicine, economics, and other disciplines.

## 1 INTRODUCTION

Causal inference is a cornerstone of empirical research in disciplines such as economics (Angrist, 1990; Imbens & Angrist, 1994), medicine (Feuerriegel et al., 2024; Weberpals et al., 2025), and marketing (Varian, 2016). It enables the estimation of causal effects from observational and randomized data, which is essential for reliable decision-making (Kern et al., 2025). In personalized medicine, for instance, it supports identifying the most effective treatment by predicting patient outcomes under different therapeutic options.

In recent years, machine learning, and especially deep learning methods, have gained significant traction in causal inference (Curth & van der Schaar, 2021; Ma et al., 2025; 2024; Melnychuk et al., 2022; Schweisthal et al., 2023; Shalit et al., 2017a; Shi et al., 2019)). These methods offer several advantages for causal effect estimation in practice, including the ability to handle large-scale and high-dimensional datasets with complex confounding structures and to model heterogeneity of causal effects (Feuerriegel et al., 2025). However, most existing approaches require retraining a model for each new dataset. To this end, existing approaches lack the flexibility to perform inference for new datasets without additional retraining, which limits their practicality in real-world settings.

---

[*]Equal contribution.
[†]Corresponding author

Meanwhile, foundation models have emerged as a transformative paradigm in machine learning (Devlin, 2018; Lahat et al., 2024; Touvron et al., 2023b;a), which offer a key advantage in that they allow for flexible, test-time inference without retraining. These models are pre-trained on large datasets and can generalize across tasks and domains. Examples include large language models (LLMs) in natural language processing and vision transformers in computer vision. However, this paradigm shift toward test-time inference has not yet had a comparable impact on causal inference. Most current approaches in causal machine learning still rely on specialized models tailored to specific tasks, requiring practitioners to manually select, train, and validate appropriate estimation methods for each new dataset.

In this paper, we propose a change to the paradigm for causal inference based on the idea of foundation models trained for tabular causal inference. For this, we build on the recently proposed prior-data fitted networks (PFNs) (Müller et al., 2022; Hollmann et al., 2023), which are transformers pre-trained on purely synthetic datasets generated from a prespecified prior distribution. PFNs enable Bayesian inference purely through in-context learning, allowing for flexible and efficient predictions without requiring additional training for new tasks (Nagler, 2023). While recent works have demonstrated the effectiveness of tabular foundation models based on PFNs for various tasks, only two concurrent works have proposed PFNs tailored for causal inference (Balazadeh et al., 2025; Robertson et al., 2025). However, these are either restricted to specific causal inference settings (namely, **only** back-door adjustment) or do **not** offer identifiability guarantees.

We introduce CausalFM, a comprehensive framework for training PFN-based foundation models for various causal inference settings. For this purpose, we introduce CausalFM priors: a novel family of prior distributions based on structural causal models respecting the underlying causal inference problem at hand. We first formalize and derive necessary criteria on how to construct such SCM-based priors for causal inference in principle. Then, we propose a concrete instantiation using Bayesian neural networks and provide a learning algorithm that leverages the SCM's ability to simulate interventional data to perform Bayesian inference in various causal inference settings.

Compared to classical causal inference methods, models trained based on our CausalFM offer the following advantages: (i) There is **no** need for additional training for new datasets as our CausalFM performs inference *entirely through in-context learning*, enabling fast and flexible deployment across new datasets. (ii) The Bayesian nature of our CausalFM provides *principled uncertainty quantification*, which is critical for downstream decision-making and for detecting situations with poor treatment overlap. (iii) The model *automatically* learns to "select" an identifiability formula based on the data distribution and task at hand. (iv) Our CausalFM builds upon rigorous identifiability guarantees to ensure valid causal inference.

Our **contributions**[1] are: (1) We formalize the constructions of priors based on structural causal models (SCMs) for Bayesian causal inference and derive necessary conditions for their validity. (2) We propose an explicit *CausalFM* prior based on Bayesian neural networks that are compatible with the structure of the causal inference problem at hand. We also propose a learning algorithm to train PFNs for causal inference problems that leverages our CausalFM prior to simulate counterfactuals to mitigate the fundamental problem of causal inference. (3) We propose concrete instantiations of our framework by training PFNs for estimating conditional average treatment effects (CATEs) in different causal inference settings. We show empirically that CausalFM performs competitively and outperforms current state-of-the-art CATE estimators on a variety of benchmarks.

## 2 RELATED WORK

We provide an overview of related literature streams. Additional related work is in Appendix A.

**Amortized causal inference.** Several recent papers pre-train large neural networks on synthetic data so that they can solve causal tasks via in-context learning. Examples include causal discovery (Mahajan et al., 2025), ATE estimation under unconfoundedness (Zhang et al., 2024), zero-shot- and few-shot learning (Nilforoshan et al., 2023; Iwata & Chikahara, 2023), and reinforcement-learning (Lee et al., 2023). These methods validate the feasibility of treating causal inference as an in-context

---

[1]Our CausalFM Toolkit is available at `https://github.com/yccm/CausalFM-toolkit`; Toolkit docs are at `https://causalfm-toolkit.readthedocs.io`; Project code is available at `https://github.com/yccm/CausalFM`.

learning problem but remain restricted to specific causal inference settings, which typically do **not** allow accommodating unobserved confounding.

Black-box causal inference (BBCI) (Bynum et al., 2025) proposes synthetically-pretrained models to perform causal inference in a variety of settings. However, their approach is different: (i) BBCI does *not* build upon a Bayesian framework. In contrast, building upon PFNs allows us to perform approximate Bayesian causal inference and thus provide rigorous uncertainty quantification. (ii) The proposed data-generating processes in BBCI are *not* tailored for high-dimensional causal inference settings (as the authors mention in their Sec. 7). In contrast, our CausalFM prior leverages Bayesian neural networks inspired by TabPFN (Hollmann et al., 2023) to create SCM-based prior distributions. (iii) Beyond proposing a new method, we provide novel formalizations and theoretical results of constructing valid SCM-based priors for Bayesian causal inference.

Finally, Dhir et al. (2025) propose a framework for amortized estimation of interventional distributions using neural processes (Garnelo et al., 2018). Our work differs in that we focus on inference on identified causal quantities without knowledge of the full causal graph (e.g., using clusters of confounders).

**PFNs for causal inference:** We are aware of only two concurrent works that propose PFN-based models for causal inference, but each with clear limitations (see Figure 1): (i) (Balazadeh et al., 2025) proposes a PFN similar to ours, but it is restricted to **only** back-door adjustment, i.e., imposes the unconfoundedness assumption throughout their paper. In

Table 1: **Overview of identifiability of PFN-based frameworks for causal inference**.

| Framework | Backdoor | Frontdoor | IV |
|---|---|---|---|
| CausalPFN Balazadeh et al. (2025) | ✓ | ✗ | ✗ |
| Do-PFN Robertson et al. (2025) | ✓ | ✗ | ✗ |
| **Ours** (CausalFM) | ✓ | ✓ | ✓ |

contrast, we propose a framework for constructing PFN-based foundation models for a *large* class of causal inference problems, *including both front-door adjustment and instrumental variable settings with unobserved confounding*. (ii) Robertson et al. (2025) proposes to train a *single* PFN on various different causal inference settings *without* providing identifiability assumptions to the model. We will show later that the approach of Robertson et al. (2025) has a crucial drawback: because the causal quantity of interest is **not** identified, the PFN learns a posterior that *may never concentrate around the true causal quantity, thus leading to asymptotically non-informative estimators*. In contrast, we propose to infuse our PFNs with identifiability assumptions required for informative causal inference. As such, we follow established philosophy in causal inference that separates identifiability and estimation steps (Kern et al., 2025; Pearl, 2009): the identifiability step should be established by the practitioner using domain knowledge (e.g., establishing whether a certain variable is a valid instrument), while the estimation step can be treated as a purely statistical learning problem.

## 3 PROBLEM SETUP

### 3.1 BACKGROUND ON PFNS

In tabular prediction problems, one considers a population $(X, Y) \sim \mathbb{P} \in \mathcal{P}$. Given a finite sample $\mathcal{D}_n \sim \mathbb{P}$ of size $n$, the goal is to estimate the conditional distribution $\mathbb{P}(Y = y \mid X = x)$. PFNs formulate this task in a Bayesian non-parametric way by placing a prior distribution $\Pi$ on $\mathcal{P}$, i.e., a *prior over data-generating distributions* (Müller et al., 2022; Nagler, 2023). Sampling proceeds hierarchically via $\mathbb{P} \sim \Pi$ and i.i.d. data $(X_i, Y_i) \sim \mathbb{P}$. Then, Bayes' rule yields the posterior distribution $\Pi(\mathbb{P} \mid \mathcal{D}_n) \propto \Pi(\mathcal{D}_n \mid \mathbb{P}) \Pi(\mathbb{P})$, where $\Pi(\mathcal{D}_n \mid \mathbb{P})$ is the likelihood of the sample $\mathcal{D}_n$ under $\mathbb{P}$ and $\propto$ denotes proportionality up to a multiplicative constant. The corresponding *posterior-predictive distribution* is the probability of $Y$ given test point $x$ and observed data $\mathcal{D}_n$, i.e.,

$$\Pi(Y \mid \mathcal{D}_n, x) = \int \mathbb{P}(Y \mid X = x) \, \Pi(\mathbb{P} \mid \mathcal{D}_n) \, d\mathbb{P}. \tag{1}$$

PFNs are neural networks $q_\theta(Y \mid \mathcal{D}_n, x)$ that parameterize the family of predictive posterior distributions with trainable parameters $\theta$. That is, PFNs map the entire dataset $\mathcal{D}_n$ and a query $x$ to a distribution over $\mathcal{Y}$. In terms of architecture, PFNs are permutation-equivariant transformers (Ashish Vaswani et al., 2017) as they allow for scalable training and leverage the attention mechanism to effectively extract information from $\mathcal{D}_n$. PFNs are trained by minimizing the negative log-likelihood loss $\mathcal{L}(\theta) = \mathbb{E}_{N \sim \Pi_N}\big[\mathbb{E}_{\mathbb{P} \sim \Pi}\big[-\log q_\theta(Y \mid X, \mathcal{D}_N)\big]\big]$, where $\Pi_N$ is a prior on the sample sizes. In

practice, we sample a sample size $N_j \sim \Pi_N$, a probability distribution $\mathbb{P}_j \sim \Pi$, a dataset $\mathcal{D}^j_{N_j} \sim \mathbb{P}_j$, and test points $(x_j, y_j) \sim \mathbb{P}_j$ and then approximate the PFN loss via

$$\hat{\mathcal{L}}(\theta) = \sum_j \left[ -\log q_\theta(y_j \mid \mathcal{D}^j_{N_j}, x_j) \right], \tag{2}$$

which is consistent for the exact posterior-predictive under regularity conditions (Nagler, 2023). Note that *all* training data are synthetic, i.e., sampled from the prior $\Pi$. Furthermore, the trained PFN can be deployed on arbitrary real datasets *without* further training.

## 3.2 TASK: CAUSAL INFERENCE

In this paper, we aim to extend PFNs to causal inference. Here, the main challenge is that the object of interest is an *interventional*[2] distribution $\mathbb{P}_{\mathrm{int}}$, yet we only observe data $\mathcal{D}_n \sim \mathbb{P}_{\mathrm{obs}}$ from a potentially different *observational* distribution (Pearl, 2009).

**Motivation.** As an illustrative example, we consider a standard causal inference setting, called *backdoor adjustment*, where the data comprise $(X, A, Y) \sim \mathbb{P}_{\mathrm{obs}}$, where $X$ are patient covariates, $A$ is a treatment, and $Y$ is an outcome of interest (van der Laan & Rubin, 2006). For example, in medicine, $X$ may contain treatment history or demographic attributes, $A$ may be a medical treatment, and $Y$ a health outcome. Following the potential outcome framework (Rubin, 1974), let $Y(a)$ denote the outcome that would be realized under the treatment $A = a$. The interventional distribution is thus over $(X, A, Y(1) - Y(0)) \sim \mathbb{P}_{\mathrm{int}}$, and a common target functional is the *conditional average treatment effect (CATE)* $Q(x) = \mathbb{E}[Y(1) - Y(0) \mid X = x]$ (Wager & Athey, 2018). The CATE quantifies the expected benefit of providing treatment given the patient's covariates.

**Identifiability.** To estimate CATE from observational data, we need to impose *identifiability assumptions*, which link the observational and the interventional distributions and allow us to express $Q$ as a functional of the observational distribution (Rosenbaum & Rubin, 1983). These are (i) consistency: $Y(A) = Y$, (ii) positivity: $\mathbb{P}_{\mathrm{obs}}(A = 1 \mid X = x) > 0$, and (iii) Unconfoundedness: $Y(1), Y(0) \perp\!\!\!\perp A \mid X$ in $\mathbb{P}_{\mathrm{int}}$.

**Generalized causal inference setting.** In the following, we provide a generalized definition of a causal inference setting, that allows us to reason about arbitrary causal inference settings and provide generalized statements beyond the standard example above.

**Definition 3.1.** We define a *causal inference setting* is a tuple $\mathcal{C} = (O, \mathcal{P}_{\mathrm{obs}} \times \mathcal{P}_{\mathrm{int}}, Q)$, where $O$ collects the observed variables (and contains at least $A$ and $Y$); $(\mathbb{P}_{\mathrm{obs}}, \mathbb{P}_{\mathrm{int}}) \in \mathcal{P}_{\mathrm{obs}} \times \mathcal{P}_{\mathrm{int}}$ are paired observational/interventional distributions over $O$ that correspond to an intervention on $A$; and $Q(\mathbb{P}_{\mathrm{int}})$ is a causal query that is *identifiable*, i.e. there exists a measurable functional $\bar{Q}$ such that $Q(\mathbb{P}_{\mathrm{int}}) = \bar{Q}(\mathbb{P}_{\mathrm{obs}})$ for all $(\mathbb{P}_{\mathrm{obs}}, \mathbb{P}_{\mathrm{int}}) \in \mathcal{P}_{\mathrm{obs}} \times \mathcal{P}_{\mathrm{int}}$.

### 3.2.1 RUNNING EXAMPLES

■ **Example 1 (back-door adjustment).** Here, we continue the example from above and define $O = (X, A, Y) \sim \mathbb{P}_{\mathrm{obs}}$ with binary $A$ as above. $\mathcal{P}_{\mathrm{obs}} \times \mathcal{P}_{\mathrm{int}}$ contains all observational and interventional distributions that satisfy consistency, positivity, and unconfoundedness. The causal query is the CATE $Q(\mathbb{P}_{\mathrm{int}})(x) = \mathbb{E}[Y(1) - Y(0) \mid X = x]$, which is identified as

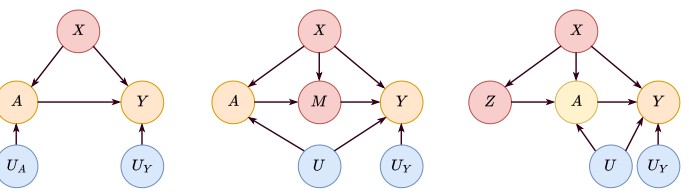

Figure 1: **C-DAGs compatible with the three example causal inference settings.** Yellow variables are observed, blue variables are unobserved, and red variables are clusters of variables.

$$\bar{Q}(\mathbb{P}_{\mathrm{obs}})(x) = \mathbb{E}_{\mathbb{P}_{\mathrm{obs}}}[Y \mid A = 1, X = x] - \mathbb{E}_{\mathbb{P}_{\mathrm{obs}}}[Y \mid A = 0, X = x]. \tag{3}$$

---

[2]Causal literature often distinguishes between interventional and counterfactual distributions. This is not relevant for the methods of our paper, and we thus use interventional distribution as an umbrella term.

■ **Example 2 (front-door adjustment).** Let $O = (X, A, M, Y) \sim \mathbb{P}_{\text{obs}}$, where $X$, $A$, and $Y$ are defined as above and $M$ is a mediator between $A$ and $Y$. Interventional distributions are defined using potential outcomes, i.e., $(X, A, M(1), M(0), Y(1, M(1)), Y(0, M(0))) \sim \mathbb{P}_{\text{int}}$, and the causal query of interest again the CATE $Q(\mathbb{P}_{\text{int}})(x) = \mathbb{E}_{\mathbb{P}_{\text{int}}}\big[Y(1, M(1)) - Y(0, M(0)) \mid X = x\big]$.

*Identifiability assumptions.* We restrict to pairs $(\mathbb{P}_{\text{obs}}, \mathbb{P}_{\text{int}})$ that satisfy (i) consistency: $Y = Y(A, M)$ and $M = M(A)$; (ii) positivity: $\mathbb{P}_{\text{obs}}(A = a \mid X = x) > 0$ and $\mathbb{P}_{\text{obs}}(M = m \mid A = a, X = x) > 0$; and (iii) front-door criterion $M(a) \perp\!\!\!\perp A \mid X = x$, and $Y(a', m) \perp\!\!\!\perp M \mid A = a', X = x$. Under these assumptions, the CATE is identified and $\widetilde{Q}$ is given via the conditional version of Pearl's front-door adjustment formula (Pearl, 2009).

■ **Example 3 (Instrumental variables).** Let $O = (X, Z, A, Y) \sim \mathbb{P}_{\text{obs}}$, where $Z$ is an instrumental variable that causes the treatment $A$ but does not directly cause the outcome $Y$. The interventional distribution is defined on $(X, Z, A, Y(1), Y(0)) \sim \mathbb{P}_{\text{int}}$ for a fixed treatment intervention $A = a$. We are again interested in the CATE $Q(\mathbb{P}_{\text{int}})(x) = \mathbb{E}\big[Y(1) - Y(0) \mid X = x\big]$.

*Identifiability assumptions.* We restrict to pairs $(\mathbb{P}_{\text{obs}}, \mathbb{P}_{\text{int}})$ that satisfy the following conditions (Newey & Powell, 2003): (i): Additive structural equation: $Y = f(X, A) + g(X, U)$, with (unknown) functions $f$ and $g$ and unobserved confounder $U$, implying that $Y$ does not directly depend on $Z$; (ii) Independence: $U \perp\!\!\!\perp Z \mid X$; (iii) Relevance: $\mathbb{P}_{\text{obs}}(A \mid X = x, Z = z) > 0$ is non-constant in $z$; and (iv) Completeness: For every measurable $g$, if $\mathbb{E}_{\mathbb{P}_{\text{obs}}}[f(x, A) \mid X = x, Z = z] = 0$ for all $z$, then $f(x, A) = 0$ almost surely in $\mathbb{P}_{\text{obs}}$. Then, the CATE can be shown to be identified via an integral equation (Newey & Powell, 2003; Hartford et al., 2017).

**Research question:** PFNs have shown to be an effective way to construct tabular foundation models. However, a causal inference setting $\mathcal{C}$ comes with additional challenges, such as the distinction between observational and interventional distribution as well as identifiability assumptions.

---

**Research question**

How can we train PFNs for a causal inference setting $\mathcal{C}$ that provides a Bayesian estimator of $Q(\mathbb{P}_{\text{int}})$ given an observational dataset $\mathcal{D}_n \sim \mathbb{P}_{\text{obs}}$ and some context (e.g., values $x$ or $a$)?

---

In the following, we introduce CausalFM consisting of (i) appropriate prior distributions that allow for approximating *interventional* predictive posterior distributions as in Eq. (2)(Section 4) and (ii) a training algorithm for the underlying PNFS (see Section 5).

## 4 CAUSALFM: PRIORS

In this section, we construct prior distributions for CausalFM which are based upon identifiable structural causal models (SCMs). We motivate and formalize our approach (Sec. 4.1, provide necessary criteria for valid causal inference (Sec. 4.2), and finally provide a method for constructing such priors in practice (Sec. 4.3). We also provide a complete toy example in Appendix B.

### 4.1 INTRODUCING SCM-BASED PRIORS

**Naïve approach.** A naïve approach for causal inference would construct a prior $\Pi$ directly for the observational distribution $\mathbb{P}_{\text{obs}}$. If the posterior $\Pi(\mathbb{P}_{\text{obs}} \mid \mathcal{D}_n) \to \mathbb{P}_{\text{obs}}^*$ converges to the ground-truth observational distribution $\mathbb{P}_{\text{obs}}^*$ (i.e., satisfying a Bernstein-von-Mises theorem), we can obtain a consistent Bayesian estimator of our causal query via $\bar{Q}(\Pi(\mathbb{P}_{\text{obs}} \mid \mathcal{D}_n))$. Accordingly, we *could* train a PFN $q_\theta(Y \mid \mathcal{D}_n, x)$ with the loss in Eq. (2) and estimate the CAPO via $\bar{Q}(q_\theta(Y \mid \mathcal{D}_n, x))$.

*However*, the above approach has *drawbacks*: (i) It requires knowledge of the identification formula $\bar{Q}$, which must be determined on a case-by-case basis depending on the causal inference setting $\mathcal{C}$ at hand. This can be tedious or even hard to compute in practice. For example, the IV setting from Example 3 requires solving integral equations to compute $\bar{Q}$ (Newey & Powell, 2003). (ii) Constructing a prior for $\mathbb{P}_{\text{obs}}$ makes it harder control the distribution of the causal query $Q$ directly. It has been shown in the literature that this can lead to prior misspecification for Bayesian causal inference or slowly converging posterior distributions (Linero & Antonelli, 2022).

**Modeling the interventional distribution.** Motivated by these drawbacks of constructing priors for only $\mathbb{P}_{\text{obs}}$, we propose to construct priors for *observational-interventional distribution pairs* $(\mathbb{P}_{\text{obs}}, \mathbb{P}_{\text{int}})$, resulting in priors defined on $\mathcal{P}_{\text{obs}} \times \mathcal{P}_{\text{int}}$. This addresses both drawbacks by (i) inducing an *interventional posterior distribution*, thus only requiring knowledge of $Q$ (not $\bar{Q}$); and (ii) we will see that priors on $\mathcal{P}_{\text{obs}} \times \mathcal{P}_{\text{int}}$ often allow to specify the prior distribution of $Q(\mathbb{P}_{\text{int}})$ directly. A natural way to define distributions on $\mathcal{P}_{\text{obs}} \times \mathcal{P}_{\text{int}}$ is via structural causal models (SCMs).

**Definition 4.1** (SCMs (Pearl, 2009)). A (semi-Markovian) *structural causal model (SCM)* $\mathcal{S}$ is a tuple $(Z, U, f, \mathbb{P})$, where $\mathbf{Z} = (Z_1, \ldots, Z_k)$ are observable **endogenous** variables, $U$ collects latent **exogenous** variables, $f = \{f_{Z_1}, \ldots, f_{Z_k}\}$ contains structural assignments $Z_i = f_{Z_i}(pa(Z_i))$ with parents $pa(Z_i) \subseteq Z \cup U$, and $\mathbb{P}$ is a joint distribution on $U$.

Every SCM induces a unique directed acyclic graph (DAG), $\mathcal{G}^{\mathcal{S}}$ by defining mapping of the parents $pa(Z_i)$ to $Z_i$ with directed edges. We distinguish two types of latent variables $U_i$ in $\mathcal{G}^{\mathcal{S}}$: $U_i$ is an *unobserved confounder* if it is the parent of both $A$ and $Y$, otherwise, we call it a *noise variable*. Intuitively, an SCM is a simulator: we can draw latent variables $U \sim \mathbb{P}$ and pass them through structural functions $f$, resulting in an induced observational distribution $\mathbb{P}_{\text{obs}}^{\mathcal{S}}$ over $Z$. At the same time, we can modify the SCM by intervening on a variable via $do(A = a)$, i.e., fixing the variable and then sampling from the SCM mechanism. This induces a corresponding interventional distribution $\mathbb{P}_{\text{int}}^{\mathcal{S}}$. We call an SCM $\mathcal{S}$ *compatible* with a causal inference setting $\mathcal{C}$, if $(\mathbb{P}_{\text{obs}}^{\mathcal{S}}, \mathbb{P}_{\text{int}}^{\mathcal{S}}) \in \mathcal{P}_{\text{obs}} \times \mathcal{P}_{\text{int}}$.

**Definition 4.2** ($\mathcal{C}$-SCM-Priors). A $\mathcal{C}$-*SCM-Prior* is any probability measure $\Pi(\mathcal{S})$ that puts all its mass on SCMs compatible with $\mathcal{C}$. Via the map $\mathcal{S} \mapsto (\mathbb{P}_{\text{obs}}^{\mathcal{S}}, \mathbb{P}_{\text{int}}^{\mathcal{S}})$ every such prior induces a distribution $\Pi\big((\mathbb{P}_{\text{obs}}, \mathbb{P}_{\text{int}})\big)$ on $\mathcal{P}_{\text{obs}} \times \mathcal{P}_{\text{int}}$.

Sampling from $\Pi$ therefore amounts to sampling a random latent distribution $\mathbb{P}$ over $U$ as well as random functional assignments $f$. These can then be used to internally sample an observational dataset $\mathcal{D}_n$, i.e., there is a well-defined likelihood $\Pi(\mathcal{D}_n \mid \mathcal{S})$ induced by $\mathcal{S}$. As a consequence, we can define the posterior distribution over SCMs via $\Pi(\mathcal{S} \mid \mathcal{D}_n) \propto \Pi(\mathcal{D}_n \mid \mathcal{S})\Pi(\mathcal{S})$, where $\propto$ denotes proportionality up to a normalization constant.

**Cluster-DAGs.** Because an SCM-prior induces a distribution over possibly many DAGs, we compress them into a shared structure. Given variables $(Z, U)$, a Cluster-DAG (C-DAG) (Anand et al., 2023)] is a DAG on clusters $C_1, \ldots, C_k$ which are disjoint subsets of $(Z, U)$. Each $\mathcal{C}$-SCM-Prior induces a unique C-DAG via the following algorithm: (i) draw an edge $C_i \rightarrow C_j$ whenever any SCMs $\mathcal{S}$ with $\Pi(\mathcal{S}) > 0$ contains some arrow from any node of $C_i$ to any node of $C_j$ *and* no SCM $\mathcal{S}$ with $\Pi(\mathcal{S}) > 0$ contains some arrow from any node of $C_j$ to any node of $C_i$; (ii) merge $C_i$ and $C_j$ whenever both directions occur across SCMs $\mathcal{S}$ with $\Pi(\mathcal{S}) > 0$.

### 4.2 Well-specified priors

The question is now how we should design our prior $\Pi$ such that the induced posterior $\Pi(\mathcal{S} \mid \mathcal{D}_n)$ allows for valid Bayesian causal inference. We now define a key desirable property of such priors. For this, we call a prior $\Pi(\mathcal{S})$ *well-specified* for $\mathcal{C}$ if, for any true pair $(\mathbb{P}_{\text{obs}}^*, \mathbb{P}_{\text{int}}^*)$ and every sample $\mathcal{D}_n \sim \mathbb{P}_{\text{obs}}^*$, it holds that

$$Q\Big(\int \mathbb{P}_{\text{int}}^{\mathcal{S}} \, \Pi(\mathcal{S} \mid \mathcal{D}_n) \, d\mathcal{S}\Big) \longrightarrow Q(\mathbb{P}_{\text{int}}^*), \quad n \to \infty. \tag{4}$$

In other words, a well-specified prior ensures that the causal query $Q$ evaluated on the *posterior-predictive interventional distribution* (PPID) $\int \mathbb{P}_{\text{int}}^{\mathcal{S}} \, \Pi(\mathcal{S} \mid \mathcal{D}_n) \, d\mathcal{S}$ is a consistent estimator of the causal target. If we were able to train a PFN to approximate the PPID of a well-specified prior, we are sure that we can apply $Q$ on this distribution and obtain a consistent estimator.

**Identifiability.** At this point, one may wonder why we only focus on priors for *identifiable* causal inference settings. Indeed, a recently proposed method, called *do-PFN* (Robertson et al., 2025), does not restrict its PFN priors to identifiable settings. The following result shows that, under weak assumptions, such priors *cannot* be well-specified, leading to asymptotic inconsistency.

**Theorem 4.3** (Identifiability rules out prior mass on violating SCMs). *Assume the causal inference setting $\mathcal{C}$ is identifiable for the causal query $Q$ with corresponding identified functional $\bar{Q}$. Let*

$$\mathcal{Z} := \Big\{ \mathcal{S} : \ \mathbb{P}_{\text{obs}}^{\mathcal{S}} \in \mathcal{P}_{\text{obs}} \ \text{and} \ Q\big(\mathbb{P}_{\text{int}}^{\mathcal{S}}\big) \neq \bar{Q}\big(\mathbb{P}_{\text{obs}}^{\mathcal{S}}\big) \Big\}$$

*denote the set of identifiability-violating SCMs, and for any $P_{\text{obs}} \in \mathcal{P}_{\text{obs}}$ define the observational equivalence class*

$$\mathcal{E}(P_{\text{obs}}) := \{\mathcal{S} : \mathbb{P}_{\text{obs}}^{\mathcal{S}} = P_{\text{obs}}\}.$$

*Assume $Q$ is a linear functional of $\mathbb{P}_{\text{int}}$ and that non-identifiability does not cancel out in prior expectation, i.e., for every $P_{\text{obs}} \in \mathcal{P}_{\text{obs}}$ with $\Pi\big(\mathcal{Z} \cap \mathcal{E}(P_{\text{obs}})\big) > 0$,*

$$\int_{\mathcal{Z} \cap \mathcal{E}(P_{\text{obs}})} Q\big(\mathbb{P}_{\text{int}}^{\mathcal{S}}\big) \, \Pi(\mathrm{d}\mathcal{S}) \;\neq\; \bar{Q}(P_{\text{obs}}) \, \Pi\big(\mathcal{Z} \cap \mathcal{E}(P_{\text{obs}})\big).$$

*Then any prior $\Pi(\mathcal{S})$ that is well-specified for $\mathcal{C}$ must satisfy $\Pi(\mathcal{Z}) = 0$.*

*Proof.* See Appendix C. □

### 4.3 Constructing SCM-based priors

**C-DAG design.** Our method for constructing priors assumes the knowledge of a well-specified C-DAG $\mathcal{G}_c$ for $\mathcal{C}$, meaning that $\mathcal{G}_c$ is induced by some well-specified $\mathcal{C}$-SCM-Prior. Such C-DAGs are usually known for most causal inference settings (see Fig. 1 for C-DAGs compatible with the settings in Examples 1–3).

One point of ambiguity is the modeling of noise variables in C-DAGs. Here, we propose a practical design rule: if $\mathcal{G}_c$ contains an unobserved confounder between $A$ and $Y$, we only add one additional noise variable to *either* $A$ or $Y$. Conversely, if $\mathcal{G}_c$ is unconfounded, we add noise parents to *both* $A$ and $Y$ (see Fig. 1). The reasoning is as follows: if $\mathcal{G}_c$ is unconfounded, we need to add noise to both $A$ and $Y$ in order to ensure not restrict ourselves to degenerate observational distributions. Conversely, any unobserved confounder $U$ induces noise into both $A$ and $Y$, thus removing the need to add noise to both. However, it is still necessary to add *one* additional noise variable to either $A$ or $Y$ since, otherwise, any unconfounded SCM compatible with $\mathcal{G}_c$ would need to be degenerate in either $A$ or $Y$. We provide a concrete toy example in Appendix B to illustrate this.

**Prior construction.** We now propose a practical algorithm to construct $\mathcal{C}$-SCM-priors. We assume that we have access to a pair $(\mathcal{G}_c, \mathcal{I})$, where $\mathcal{G}_c$ is a well-specified C-DAG for $\mathcal{C}$ and $\mathcal{I}$ is a set of constraints on SCMs $\mathcal{S}$ compatible with $\mathcal{G}_c$ ensuring that $\mathcal{S}$ is also compatible with $\mathcal{C}$.

■ *Example 1: back-door adjustment.* The observable variables are $(X, A, Y)$ together with noise variables. A compatible C-DAG is in Fig. 1 (left). The constraint set is $\mathcal{I}(\mathcal{S}) = \{\mathbb{P}_{\text{obs}}^{\mathcal{S}}(A = a \mid X = x) > 0\}$, ensuring that all SCMs satisfy the positivity assumption.

■ *Example 2: Front-door adjustment.* Here, the observed variables are $(X, A, M, Y)$ with noise variables and an unobserved confounder $U$ between $A$ and $Y$. A compatible C-DAG is in Fig. 1 (middle). The constraint set is $\mathcal{I}(\mathcal{S}) = \{\mathbb{P}_{\text{obs}}^{\mathcal{S}}(A = a \mid X = x) > 0, \mathbb{P}_{\text{obs}}^{\mathcal{S}}(M = m \mid X = x, A = a) > 0\}$, ensuring positivity for both treatments and mediators.

■ *Example 3: Instrumental variables.* The observed variables are $(X, Z, A, Y)$, augmented by noise variables and an unobserved confounder $U$ that is a joint parent of $A$ and $Y$ has no edge to the instrument $Z$; see the compatible C-DAG in Fig. 1 (right). The constraint set is $\mathcal{I}(\mathcal{S}) = \{\mathbb{P}_{\text{obs}}^{\mathcal{S}}(Z = z \mid X = x) > 0, f_Y^{\mathcal{S}}(X, A, U) = f^{\mathcal{S}}(X, A) + g^{\mathcal{S}}(X, U)\}$.

**Overall algorithm.** Given $(\mathcal{G}_c, \mathcal{I})$, we propose to construct a prior distribution $\Pi$ over SCMs as follows: First, we order the clusters $(C_1, \ldots, C_k)$ according to their hierarchy in the DAG (i.e., $C_1$ has no parents). Then, we iterate over each cluster $C_i$ as follows: if $C_i$ only contains latent variables, fix their distribution to a standard normal distribution via $U^{(i)} \sim \mathcal{N}(0, \mathbf{I})$. If $C_i$ is a cluster of observed and latent variables, we assign a clustered Bayesian neural network (BNN) prior to $C_i$ (see below). If $C_i$ only contains observed variables, we assign an observational BNN prior.

**Clustered BNN prior.** For clusters that contain both observed and latent variables, we leverage a BNN-based prior inspired by TabPFN (Hollmann et al., 2023). This prior allows us to effectively sample potentially high-dimensional clusters of variables for which the internal causal structure is irrelevant to infer the causal query of interest. The prior is defined via

$$g_\theta^{(i)} : \text{pa}(C_i) \longrightarrow \mathbb{R}^r, \qquad \theta \sim \Pi_{C_i} \quad \text{s.t. } g_\theta^{(i)} \text{ satisfying } \mathcal{I}(\mathcal{S}_\theta). \tag{5}$$

We then sample random nodes from $g_\theta^{(i)}$ that coincide with observed nodes in $C_i$, while the remaining nodes serve as latent noise within the cluster. This corresponds to applying the approach taking in TabPFN (Hollmann et al., 2023) to clusters $C_i$ in the C-DAG in which the causal structure does not matter for estimating our causal query.

**Observational BNN prior.** If $C_i$ contains only observed nodes, we define another BNN via

$$f_\theta^{(i)} : \operatorname{pa}(C_i) \longrightarrow \mathbb{R}^{|C_i|}, \qquad \theta \sim \Pi_{C_i} \quad \text{subject to } f_\theta^{(i)} \text{ satisfying } \mathcal{I}(\mathcal{S}_\theta), \tag{6}$$

and set $C_i = f_\theta^{(i)}\big(\operatorname{pa}(C_i)\big)$. The observed nodes within $C_i$ thus correspond to the output of the neural network and are *not* randomly subsampled neurons.

*Example 1: back-door adjustment.* Here, the data distribution $\mathbb{P}$ can be separated as follows: $(X, U_X) \sim \mathbb{P}_X$ with $U_X$ denoting noise variables withing the cluster $X$, $U_A \sim \mathbb{P}_{U_A}$, $U_Y \sim \mathbb{P}_{U_Y}$, $A = f_A(X, U_A)$, and $Y = f_A(X, A, U_Y)$. Our algorithm proceeds as follows: $\mathbb{P}_{U_A}$ and $\mathbb{P}_{U_Y}$ are noise variables and are set to standard normal distributions. The cluster $(X, U_X)$ contains both noise and observed variables, meaning that $\mathbb{P}_X$ is sampled from an clustered BNN prior. Finally, $A$ and $Y$ are observed variables meaning that $f_A$ and $f_Y$ are sampled from observational BNN priors. We refer to Appendix D.1 for full implementation details, including for Example 2 and 3.

**Notes on identifiability.** Our framework follows established causal inference philosophy and separates identifiability from estimation (Pearl, 2009): the identifiability step (=choosing the causal setting) requires careful modeling and usage of domain knowledge, while the estimation step can be handed over to our CausalFM. If practitioners suspect identifiability assumptions may be violated, we recommend performing causal sensitivity analysis (Dorn & Guo, 2022; Frauen et al., 2023) to assess the extent of potential violations.

## 5 CAUSALFM: TRAINING

### 5.1 TRAINING ALGORITHM

We look at the case where the causal query $Q(\mathbb{P}_{\text{int}}(Y \mid X))$ is a function of the conditional interventional distribution $\mathbb{P}_{\text{int}}(Y \mid X)$ for some contextual observed variables $X$. This includes, e.g., the CATE $\mathbb{E}[Y(1) - Y(0) \mid X]$ and CAPO $\mathbb{E}[Y(a) \mid X]$ from our running examples.

Our goal is to train a PFN $q_\theta(Y \mid x)$ to approximate the conditional PPID (posterior predictive interventional distribution) $\Pi_{\text{int}}(Y \mid \mathcal{D}_n, X = x) = \int \mathbb{P}_{\text{int}}^{\mathcal{S}}(Y \mid X = x) \, \Pi(\mathcal{S} \mid \mathcal{D}_n) \, d\mathcal{S}$. Given an SCM prior $\Pi$ and a prior $\Pi_N$ over sample sizes, we propose the following modified PFN loss

$$\mathcal{L}(\theta) = \mathbb{E}_{N \sim \Pi_N}\big[\mathbb{E}_{\mathcal{S} \sim \Pi}\big[\mathbb{E}_{(X,Y) \sim \mathbb{P}_{\text{int}}^{\mathcal{S}}}\big[\mathbb{E}_{\mathcal{D} \sim \mathbb{P}_{\text{obs}}^{\mathcal{S}}}\big[-\log q_\theta(Y \mid X, \mathcal{D}_N)\big]\big]\big]\big]. \tag{7}$$

Importantly, the dataset $\mathcal{D}$ is sampled from the *observational* distribution, while the pair $(X, Y)$ is sampled from the *interventional* distribution induced by a random SCM. This ensures that the PFN will aim to predict the interventional outcome $Y$ based on data following the observational distribution. A similar loss has been proposed by Bynum et al. (2025), which, however, is only based on the mean-squared error instead of the negative log-likelihood and thus does not allow an interpretation for approximating the PPID in a Bayesian setting. In particular, modeling the entire PPID allows us not only to provide point estimators of our causal query, but also to account for uncertainty.

In practice, we sample the sample size $N_j \sim \Pi_N$, an SCM $\mathcal{S}_j \sim \Pi$, and an observational dataset $\mathcal{D}_{N_j}^j \sim \mathbb{P}_{\text{obs}}^{\mathcal{S}_j}$ by sampling from the SCM. Then, we modify the SCM by performing the intervention of interest (e.g., $\operatorname{do}(A = a)$) and sample test points $(x_j, y_j) \sim \mathbb{P}_{\text{int}}^{\mathcal{S}_j}$ from the interventional SCM. The approximated PFN-loss is then

$$\hat{\mathcal{L}}(\theta) = \sum_j \big[-\log q_\theta(y_j \mid \mathcal{D}_{N_j}^j, x_j)\big]. \tag{8}$$

Finally, once $q_\theta(Y \mid x)$ is trained, we can obtain an estimator for the causal query via $Q(q_\theta(Y \mid X))$, i.e., by applying the causal query on the approximated PPID by the PFN.

**Example: back-door adjustment.** Here, we sample a sample size $N_j \sim \Pi_N$, an SCM $\mathcal{S}_j \sim \Pi$ from our constructed prior distribution $\Pi$ and an observational dataset $\mathcal{D}_{N_j}^j \sim \mathbb{P}_{\text{obs}}^{\mathcal{S}_j}$. Then, we perform two interventions $do(A = 1)$ and $do(A = 0)$ to obtain test points $(x_j, y_j(1) - y_j(0)) \sim \mathbb{P}_{\text{int}}^{\mathcal{S}_j}$. The PFN loss becomes

$$\hat{\mathcal{L}}(\theta) = \sum_j \left[ -\log q_\theta(y_j(1) - y_j(0) \mid \mathcal{D}_{N_j}^j, x_j) \right]. \tag{9}$$

**Implementation details.** Each observation is tokenized during embedding, with separate encoders applied to observational variables. The resulting tokens are processed by a transformer-based PFN to obtain representations, which are subsequently passed to a Gaussian mixture model (GMM) head. Our implementation of $q_\theta(Y \mid x)$ is based on the TabPFN architecture (Hollmann et al., 2023). We train the model with a learning rate of $1\text{e}{-3}$, weight decay $1\text{e}{-5}$, batch size 16, and sequence length 1024 for up to 150 epochs. Training CausalFM on a single NVIDIA A100 GPU takes about 24 hours. Details on the data prior and generation details are provided in Appendix D.1, while the full implementation is given in Appendix D.2.

## 6 EXPERIMENTS

We evaluate our method across three causal inference settings: standard CATE estimation, instrumental variables (IV), and front-door adjustment.

**Evaluation metrics.** We report the precision in estimating heterogeneous effects (PEHE) (Curth & van der Schaar, 2021; Hill, 2011), defined as the root mean squared deviation between predicted and ground-truth CATE, to evaluate the model performance on the CATE estimation task.

Table 2: **Standard CATE estimation** over 10 synthetic datasets and Jobs dataset.

| Method | Synthetic | Jobs |
|---|---|---|
| BASELINES (A): STANDARD CATE ESTIMATORS | | |
| S-learner (Künzel et al., 2019) | $0.734 \pm 0.16$ | $0.697 \pm 0.18$ |
| T-learner (Künzel et al., 2019) | $0.661 \pm 0.17$ | $0.822 \pm 0.18$ |
| TARNet (Shalit et al., 2017b) | $0.854 \pm 0.23$ | $0.864 \pm 0.24$ |
| DR-learner (Kennedy, 2023b) | $0.765 \pm 0.17$ | $0.959 \pm 0.18$ |
| RA-learner (Curth & van der Schaar, 2021) | $0.609 \pm 0.13$ | $0.652 \pm 0.15$ |
| X-learner (Künzel et al., 2019) | $0.563 \pm 0.15$ | $0.802 \pm 0.18$ |
| BASELINES (B): FOUNDATION MODELS-BASED METHODS | | |
| CausalPFN (Balazadeh et al., 2025) | $0.557 \pm 0.18$ | $0.528 \pm 0.16$ |
| DoPFN (Robertson et al., 2025) | $0.586 \pm 0.19$ | $0.482 \pm 0.20$ |
| **CausalFM (ours)** | $0.515 \pm 0.20$ | $0.478 \pm 0.18$ |

Lower = better. Reported: PEHE (mean $\pm$ std). Top-three per column are in blue, purple, orange.

### 6.1 EVALUATION FOR STANDARD CATE SETTING

**Baselines for standard CATE estimation.** We consider a broad range of state-of-the-art methods for the conditional treatment effect estimation from the literature: (1) **S-learner** (Künzel et al., 2019): the S-learner is a model-agnostic learner that trains a single regression model by concatenating the covariate and the treatment as input; (2) **T-learner** (Künzel et al., 2019): the T-learner is a model-agnostic learner that trains separate regression models for treated and control groups; (3) **X-learner** (Künzel et al., 2019): builds upon the T-learner by first imputing individual treatment effects in each group and then fitting models to these pseudo-effects; (4) **TARNet** (Shalit et al., 2017b): using representation learning to extract features of covariates and train separate branches for treated and control groups with regularization; (5) **DR-learner** (Kennedy, 2023b): generates pseudo-outcomes based on the doubly-robust AIPW estimator; (6) **RA-learner** (Curth & van der Schaar, 2021): uses a regression-adjusted pseudo-outcome in the second stage. We also include two PFN-based foundation models for treatment effect estimation: (7) **CausalPFN** (Balazadeh et al., 2025) and (8) **DoPFN** (Robertson et al., 2025). Further implementation details are in Appendix D.3.

**Results on standard CATE estimation.** We benchmark our model on ten synthetic datasets generated under diverse mechanisms, with implementation details in Appendix D. In addition, we evaluate on a semi-synthetic version of the Jobs dataset (Smith & Todd, 2005), derived from the widely used LaLonde study (LaLonde, 1986). Here, we generate outcomes to create a semi-synthetic dataset and allow for evaluation against ground-truth.

Table 2 reports the averaged PEHE across the synthetic datasets (full results in the Appendix) and the Jobs dataset. Our experiments show that CausalFM achieves competitive CATE estimation performance across all benchmarks, *without requiring model retraining*.

**Baselines for IV setting.** We benchmark against a broad set of state-of-the-art IV methods for treatment effect estimation: (1) **KIV** (Singh et al., 2019): a nonlinear extension of two-stage least squares using kernel ridge regression with feature maps; (2) **DFIV** (Xu et al., 2021): extends KIV by parameterizing feature maps with neural networks trained iteratively; (3) **DeepIV** (Hartford et al., 2017): a two-stage neural approach, first estimating the treatment distribution and then solving a counterfactual prediction task; (4) **DeepGMM** (Bennett et al., 2019): formulates IV estimation as a minimax game based on the generalized method of moments, solved via adversarial training; (5) **DMLIV** (Syrgkanis et al.,

Table 3: **IV setting** for CATE estimation with binary and continuous instrument variables.

| Method | Binary IV | Continuous IV |
|---|---|---|
| BASELINES (A): STANDARD IV ESTIMATORS | | |
| KIV (Singh et al., 2019) | $0.454 \pm 0.16$ | $0.577 \pm 0.20$ |
| DRIV (Syrgkanis et al., 2019) | $0.531 \pm 0.18$ | $0.693 \pm 0.20$ |
| DeepIV (Hartford et al., 2017) | $0.427 \pm 0.15$ | $0.516 \pm 0.13$ |
| DeepGMM (Bennett et al., 2019) | $0.503 \pm 0.20$ | $0.588 \pm 0.21$ |
| DMLIV (Syrgkanis et al., 2019) | $0.479 \pm 0.23$ | $0.618 \pm 0.20$ |
| DFIV (Xu et al., 2021) | $0.709 \pm 0.29$ | $0.583 \pm 0.30$ |
| MRIV (Frauen & Feuerriegel, 2022) | $0.688 \pm 0.21$ | $0.641 \pm 0.24$ |
| BASELINES (B): FOUNDATION MODELS-BASED METHODS | | |
| DoPFN (Robertson et al., 2025) | $0.523 \pm 0.20$ | $0.675 \pm 0.37$ |
| **CausalFM (ours)** | $0.422 \pm 0.16$ | $0.579 \pm 0.21$ |

Lower = better. Reported: PEHE (mean $\pm$ standard deviation). Top-three per column are in blue, purple, orange.

2019): a double machine learning framework that estimates nuisance functions and learns the CATE by orthogonalized regression; (6) **DRIV** (Syrgkanis et al., 2019): a meta-learner combining DMLIV with doubly robust pseudo-outcomes for improved stability; and (7) **MRIV** (Frauen & Feuerriegel, 2022): a multiply robust framework for binary IVs that directly estimates CATE via pseudo–outcome regression. For foundation model baselines, as CausalPFN (Balazadeh et al., 2025) is only for back-door adjustment, we include DoPFN (Robertson et al., 2025).

## 6.2 EVALUATION FOR IV SETTING

**Results on IV setting.** We evaluate our models on datasets with varying confounding strengths. Table 3 reports the averaged PEHE for binary and continuous IVs. Note that CausalPFN is *not* designed for IV settings. In contrast, we find that our CausalFM consistently achieves comparable performance relative to standard IV estimators and outperforms biased alternatives. Importantly, in contrast the the standard baselines, these results hold *without requiring model retraining*. Hence, this confirms the flexibility of our approach to IV settings.

Table 4: **Front-door adjustment setting** for CATE estimation.

| Method | PEHE |
|---|---|
| BASELINES (A): STANDARD FRONT DOOR ADJUSTMENT | |
| Plug-in front-door learner (Linear) Pearl (2009) | $1.124 \pm 0.28$ |
| Plug-in front-door learner (RF) Pearl (2009) | $1.364 \pm 0.52$ |
| Plug-in front-door learner (NN) Pearl (2009) | $0.889 \pm 0.38$ |
| BASELINES (B): FOUNDATION MODELS-BASED METHODS | |
| DoPFN (Robertson et al., 2025) | $1.274 \pm 0.24$ |
| **CausalFM (ours)** | $0.847 \pm 0.34$ |

Lower = better. Reported: PEHE (mean $\pm$ standard deviation). Top-three per column are in blue, purple, orange.

## 6.3 FRONT-DOOR ADJUSTMENT

We additionally evaluate our model under the front-door adjustment setting in Table 4. Due to space constraints, details are provided in Appendix H.1. The experiments show the flexibility of our method to perform causal inference in the front-door adjustment setting.

## 6.4 DISCUSSION

**Limitations and future work.** The current evaluation is limited to synthetic and semi-synthetic data due to the fundamental problem of missing potential outcomes on real-world data. For future work, it will be interesting to investigate the performance of CausalFM in applied A/B experimental setups to assess its empirical performance and robustness under real-world conditions. Additionally, an important research direction will be to incorporate interpretability or fairness constraints into CausalFM, which is crucial for reliable deployment in practice.

**Ethics statement.**

*Human subjects and IRB.* This work does not involve experiments with human subjects. Our training data are *synthetically* generated from prespecified SCM-based priors. For empirical evaluation, we additionally use publicly available benchmark data (e.g., Jobs) where outcomes are generated in a semi-synthetic manner following common practice; no identifiable personal information is introduced by us. Accordingly, no IRB review was required for this study.

*Data, privacy, and security.* We do not collect, store, or release sensitive personal data. Public datasets are used under their respective licenses, and our semi-synthetic outcome generation avoids re-identification risks. We will document preprocessing and generation steps to support reproducibility.

*Bias and potential harms.* Causal estimators can be misused if applied outside the assumed identification regime (e.g., back-door, front-door, IV) or under severe violations (e.g., weak instruments, lack of overlap). To mitigate harm: (i) we make assumptions explicit and provide uncertainty quantification; (ii) we advocate domain-expert validation and sensitivity checks before deployment; (iii) we discourage high-stakes automated decision-making without human oversight.

*Use of large language models (LLMs).* We used LLM-based tools to assist with writing (clarity, grammar) and for literature research. All claims were authored and verified by the authors; citations were cross-checked against primary sources. No sensitive data were provided to LLM tools.

**Reproducibility statement.** We ensure reproducibility of our results by providing the full implementation and training scripts. Our CausalFM Toolkit is available at `https://github.com/yccm/CausalFM-toolkit`; Toolkit docs are at `https://causalfm-toolkit.readthedocs.io`; Project code is available at `https://github.com/yccm/CausalFM`. The repository contains the necessary code to reproduce our experiments, along with instructions for dataset preparation, model training and evaluation procedures. This setup allows independent researchers to replicate the reported results and extend our work with minimal effort.

**Acknowledgments.** This paper is supported by the DAAD program "Konrad Zuse Schools of Excellence in Artificial Intelligence", sponsored by the Federal Ministry of Research, Technology, and Space. Additionally, this work has been supported by the German Federal Ministry of Education and Research (Grant: 01IS24082).

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

# A    EXTENDED RELATED WORK

**Prior-data-fitted networks (PFNs) as tabular foundation models.**    Foundation models are an emerging paradigm that has revolutionized machine learning for various data modalities, particularly language and vision tasks (Devlin, 2018; Lahat et al., 2024; Touvron et al., 2023b;a). The same paradigm is now being explored for tabular data – a modality that underpins the large majority of analyses in science and business (van Breugel & van der Schaar, 2024). Prior-data-fitted networks (PFNs) (Müller et al., 2022) constitute a powerful approach to training tabular foundation models. PFNs are large transformers trained on synthetic data to perform Bayesian inference through in-context learning. TabPFN (Hollmann et al., 2023; 2025) scaled this idea by pairing the transformer with a Bayesian neural network prior over structural causal models (SCMs) and demonstrating state-of-the-art performance on various tabular benchmarks. Subsequent work extended PFNs to time–series forecasting (Hoo et al., 2025) and analyzed their in-context learning abilities theoretically (Nagler, 2023). Critically, all existing PFNs are trained *only* for *predictive* tasks and do **not** target causal estimands; they therefore are **not** designed for causal inference of treatment effects, which is the goal of our paper.

**Treatment effect estimation.**    Causal inference, such as the estimation of average treatment effects, originates from fields like econometrics (Imbens & Angrist, 1994; Angrist, 1990), statistics (van der Laan & Rubin, 2006), and epidemiology (Robins, 1986; 1994). Machine learning methods have been proposed to estimate *heterogeneous* effects to support personalized decision-making. One line of work are frequentist methods, which often build on semiparametric theory (Robins et al., 1994; Robins, 1999), yielding model-agnostic estimators that are doubly robust and Neyman-orthogonal (van der Laan, 2006; Chernozhukov et al., 2018; Nie & Wager, 2021; Foster & Syrgkanis, 2023; Kennedy, 2023a). Another line of work builds upon specific machine learning methods/ architectures such as regression trees (Wager & Athey, 2018) or neural networks (Johansson et al., 2016; Shalit et al., 2017a; Shi et al., 2019) and adopts them to causal inference. Bayesian alternatives include Bayesian additive regression trees (Hahn et al., 2020) or Gaussian-process counterfactual regression (Alaa & van der Schaar, 2017). However, all of the existing estimators above must be *retrained* for every new dataset. In contrast, our CausalFM allows for pre-trained models to approximate Bayesian causal inference.

## A.1    DIFFERENCES BETWEEN CAUSALFM, CAUSALPFN, AND DO-PFN

**Identifiability.** CausalFM separates identifiability from estimation. The central motivation of CausalFM is that causal identification (choosing an identifiable setting such as back-door, IV, or front-door) must be handled before estimation. This mirrors classical causal inference practice and ensures that the PFN *only* learns within an identifiable causal setting. Our reasoning for identifiability is as follows:

(1) Asymptotically unbiased causal inference: As we show in Theorem 4.3, incorporating identifiability assumptions into the prior is *necessary* for asymptotically unbiased causal inference. As a consequence, methods that ignore identifiability assumptions yield biased causal effect estimates, even if we collect large amounts of data. This is highly undesirable in practice.

(2) Informative predictive-posterior distributions: If we do not impose any assumption on the DGP, it is well known that causal effect estimation is not just fundamentally biased, but also that this bias can be of arbitrary size. For example, the backdoor-adjustment bias due to omitted unobserved confounding can be written in closed form depending on confounding strength. Thus, if the PFN-prior assigns positive probability mass for DGPs with arbitrary confounding strength, the predictive-posterior must respect the possibility of arbitrarily biased treatment effects, thus rendering PFN-based inference completely noninformative.

(3) Clear separation between domain knowledge and statistical inference: One might argue that a possible remedy would be to restrict the PFN prior only to DGPs with somewhat "weak" identifiability violations (e.g., weak unobserved confounding). However, we argue that this would correspond to assumptions/ domain knowledge on the DGP, similar to those in our paper, that must be made transparent for practitioners and could also possibly be violated.

In short, our paper follows established causal inference philosophy and separates identifiability from estimation: the identifiability step (choosing the causal setting) requires careful modeling and usage of domain knowledge, while the estimation step can be handed over to our CausalFM. If practitioners suspect identifiability assumptions may be violated, we recommend performing causal sensitivity analysis to assess the extent of potential violations.

In contrast, CausalPFN implicitly assumes *only* back-door adjustment and therefore *cannot* handle IV or front-door, resulting in bias under unobserved confounding. Do-PFN mixes many causal graphs in a single prior without conditioning on which setting is identifiable, which, as our Theorem 4.3 shows, can lead to asymptotically *biased* estimates and non-informative posteriors.

**Prior construction.** This philosophy requires a fundamentally different prior construction. Because identifiability is encoded at the level of causal structure, CausalFM introduces C-SCM priors and C-DAGs that enforce the assumptions required by each identifiability strategy. Do-PFN does *not* encode identifiability constraints into the prior family for their prior construction. CausalPFN is *restricted* solely to back-door adjustment.

**Theoretical guarantees.** Beyond this framework, we contribute *new* theoretical results showing that identifiability must be incorporated into PFN priors. Theorem 4.3 proves that if a PFN prior places nonzero mass on SCMs that violate the identifiability conditions of the chosen setting, then the resulting posterior predictive interventional distribution is necessarily misspecified and cannot yield consistent causal effect estimates, even with infinite data. This explains the empirical behavior of Do-PFN, which may return non-informative posteriors when its prior includes SCMs with strong unobserved confounding and no valid instruments. Our theoretical results show that this issue is structural, not merely an implementation detail.

**Empirical performance.** Empirically, our model outperforms others in different settings. Besides, we also have experiments showing the necessity to have the correct identifiability assumption in the prior specification.

# B  EXAMPLE FOR SCM-PRIORS

Here, we consider the IV setting from Example 3 with additional normality assumption and empty $X = \emptyset$, i.e., observational distribution $(Z, A, Y) \sim \mathbb{P}_{\text{obs}}$. Let us consider the following class of SCMs:

$$U \sim \mathcal{N}(0,1),\ \epsilon_Z \sim \mathcal{N}(0,1),\ \epsilon_A \sim \mathcal{N}(0,1),\ \epsilon_Y \sim \mathcal{N}(0,1), \tag{10}$$

$$Z = \alpha\epsilon_Z, +\kappa U\ A = \beta Z + \delta\epsilon_A + \gamma U,\ Y = \zeta A + \eta U + \theta\epsilon_Y, \tag{11}$$

where $U$ is an unobserved confounder between $A$ and $Y$, and $\epsilon_Z$, $\epsilon_A$, $\epsilon_Y$ are noise variables, and $\alpha$, $\beta$, $\gamma$, $\delta$, $\zeta$, $\eta$, and $\theta$ are scalars describing the functional dependences between observed and noise variables. Our causal query is $Q(\mathbb{P}_{\text{int}}) = \mathbb{E}[Y(1)] = \zeta$.

**General approach.** The class of SCMs above is compatible with the linear IV setting whenever it holds that $\kappa = 0$ (independence assumption from Example 3). Hence, we can specify a prior distribution over this class of SCMs by specifying a distribution $\Pi$ over $(\alpha, \beta, \gamma, \delta, \zeta, \eta, \theta)$ and setting $\kappa = 0$. Note that this automatically specifies a distribution over $\mathbb{P}_{\text{obs}}$ (by sampling from the SCM) *and* $\mathbb{P}_{\text{int}}$ (by intervening and setting $A = 1$ in the SCM). Interestingly, this addresses the two drawbacks of observational priors from above as follows: (i) During the PFN training we can sample $\mathcal{D}_n \sim \mathbb{P}_{\text{obs}}$ and $y(1) \sim \mathbb{P}_{\text{int}}$ and thus fit $q_\theta(y(1) \mid \mathcal{D}_n)$ for the interventional outcome (see Sec. 5 for details). For estimating the causal query we can thus use $Q(q_\theta(y(1) \mid \mathcal{D}_n))$ and do not need access to the potentially unknown $\bar{Q}$. (ii) We can directly control the marginal prior distribution of $\zeta$, thus remedying the above drawbacks and allowing us more control to incorporate prior information of our causal query.

**Adding identifiability assumptions to the prior.** A key question is whether we should actually impose the identifiability assumption $\kappa = 0$ when constructing a prior. A different approach would be to also put a prior on $\kappa$, thus taking account the possibility of identifiability violations in the prior. Such an approach has been proposed by (Robertson et al., 2025), where the authors construct a prior over many possible causal inference settings simultaneously. *However*, as we show in the following, this would make consistent Bayesian estimation of the causal query of interest *impossible*, confirming Theorem 4.3.

**Lemma B.1.** *Let $S^* = (\alpha^*, \beta^*, \delta^*, \gamma^*, \zeta^*, \eta^*, \theta^*, \kappa^* = 0)$ be an identified ground-truth SCM. Then for any causal target $\zeta \neq \zeta^*$ there exists another SCM $S = (\alpha, \beta, \gamma, \delta, \zeta, \eta, \theta, \kappa)$ with $\kappa \neq 0$ that induces the same observational distribution as $S^*$.*

*Proof.* See Appendix C. $\qquad\square$

Lemma B.1 has an important consequence: if our prior $\Pi$ puts positive probability mass on all possible combinations of $(\alpha, \beta, \gamma, \delta, \zeta, \eta, \theta, \kappa)$, the corresponding posterior $\Pi(\cdot \mid \mathcal{D}_n)$ will even for $n \to \infty$ put positive probability mass on any $\zeta \in \mathbb{R}$, thus being completely non-informative about the causal target quantity. As a consequence, any Bayesian point estimator using such a prior (e.g., as in under the approach (Robertson et al., 2025)) will be asymptotically biased.

In contrast, we present a different approach to circumvent the above problems: namely, we propose to *construct PFN-priors that incorporate assumptions that allow for identifiability of the causal target quantity* (e.g., setting $\kappa = 0$ in the above example). As such, we follow established philosophy in causal inference that separates identifiability and estimation steps (Pearl, 2009): the identifiability step should be established by the practitioner using domain knowledge (e.g., establishing whether a certain variable is a valid instrument). Once identifiability has been established, we can use Bayesian modeling and PFN-based models for the *estimation step*.

**Which noise variables to model?** A key question that remains is what classes of SCMs we can use to specify priors for the causal inference setting $\mathcal{C}$ at hand. Indeed, the class of SCMs is non-unique: as suggested in the main paper, it is not necessary to specify both noise variables $\epsilon_A$ and $\epsilon_Y$.

**Lemma B.2.** *Let $\mathcal{S}^* = (\alpha^*, \beta^*, \gamma^*, \delta^*, \zeta^*, \eta^*, \theta^*)$ be a fixed SCM from the above class with $\text{Var}^*(A \mid z) > 0$ and $\text{Var}^*(Y \mid a) > 0$ for all $z, a$. Then, there exist unique SCMs $\mathcal{S}_1 = (\alpha_1, \beta_1, \gamma_1, \delta_1 = 0, \zeta_1, \eta_1, \theta_1)$ and $\mathcal{S}_2 = (\alpha_2, \beta_2, \gamma_2, \delta_2, \zeta_2, \eta_2, \theta_2 = 0)$ that induce the same observational distribution as $\mathcal{S}^*$ and thus the same causal query $\zeta_1 = \zeta_2 = \zeta^*$. However, whenever it holds that both $\delta = 0$ and $\theta = 0$, there exists an SCM $\mathcal{S}^*$ for which $\zeta \neq \zeta^*$.*

*Proof.* See Appendix C. □

Lemma B.2 implies that it *suffices to specify priors over SCMs without either treatment noise $\epsilon_A$ or outcome noise $\epsilon_Y$*. However, if we remove both, there exist interventional distributions for which the prior will never put probability mass on the ground-truth causal query, rendering Bayesian inference inconsistent. In the following, we generalize this result to arbitrary SCMs and causal inference settings.

## C  PROOFS

### C.1  PROOF OF THEOREM 4.3

*Proof of Theorem 4.3.* Assume, for contradiction, that $\Pi(\mathcal{Z}) > 0$. Then, there exists an observational distribution $P_{\mathrm{obs}} \in \mathcal{P}_{\mathrm{obs}}$ such that

$$w := \Pi\big(\mathcal{Z} \cap \mathcal{E}(P_{\mathrm{obs}})\big) > 0, \qquad \mathcal{E}(P_{\mathrm{obs}}) := \{\mathcal{S} : \mathbb{P}_{\mathrm{obs}}^{\mathcal{S}} = P_{\mathrm{obs}}\}. \tag{12}$$

Let $\mathcal{W} := \mathcal{E}(P_{\mathrm{obs}}) \setminus \mathcal{Z}$ denote the non-violating subset of the same observational equivalence class. Pick any $\mathcal{S}^w \in \mathcal{W}$ with $\Pi(\mathcal{S}^w) > 0$ (such an SCM exists if $\Pi$ is well-specified for $\mathcal{C}$ on $P_{\mathrm{obs}}$), and draw data $\mathcal{D}_n \sim \mathbb{P}_{\mathrm{obs}}^{\mathcal{S}^w} = P_{\mathrm{obs}}$.

Consider the posterior-predictive interventional functional

$$T_n := Q\Big(\int \mathbb{P}_{\mathrm{int}}^{\mathcal{S}} \, \Pi(\mathcal{S} \mid \mathcal{D}_n) \, \mathrm{d}\mathcal{S}\Big). \tag{13}$$

Because $Q$ is linear, we have

$$T_n = \int Q(\mathbb{P}_{\mathrm{int}}^{\mathcal{S}}) \, \Pi(\mathcal{S} \mid \mathcal{D}_n) \, \mathrm{d}\mathcal{S}. \tag{14}$$

**Step 1: Split the integral.**  Write $\mathcal{R} := \mathcal{E}(P_{\mathrm{obs}})^c$. Then

$$T_n = \int_{\mathcal{E}(P_{\mathrm{obs}})} Q(\mathbb{P}_{\mathrm{int}}^{\mathcal{S}}) \, \Pi(\mathcal{S} \mid \mathcal{D}_n) \, \mathrm{d}\mathcal{S} + \int_{\mathcal{R}} Q(\mathbb{P}_{\mathrm{int}}^{\mathcal{S}}) \, \Pi(\mathcal{S} \mid \mathcal{D}_n) \, \mathrm{d}\mathcal{S}. \tag{15}$$

**Step 2: Posterior concentrates on the observational equivalence class.**  Under standard Bayesian consistency conditions for the observational model (e.g., KL support of $\Pi$ at $P_{\mathrm{obs}}$), we have

$$\Pi\big(\mathcal{R} \mid \mathcal{D}_n\big) \to 0, \qquad n \to \infty, \tag{16}$$

and, hence, the last integral over $\mathcal{R}$ vanishes asymptotically (assuming the usual integrability for $Q$).

**Step 3: Inside $\mathcal{E}(P_{\mathrm{obs}})$, the posterior is proportional to the prior.**  For every $\mathcal{S} \in \mathcal{E}(P_{\mathrm{obs}})$, the observational likelihood is identical because $\mathbb{P}_{\mathrm{obs}}^{\mathcal{S}} = P_{\mathrm{obs}}$. Therefore, for any measurable $A \subseteq \mathcal{E}(P_{\mathrm{obs}})$ and all $n$, we yield

$$\Pi(A \mid \mathcal{D}_n) = \frac{\Pi(A)}{\Pi(\mathcal{E}(P_{\mathrm{obs}}))} \qquad \text{and} \qquad \Pi(\mathrm{d}\mathcal{S} \mid \mathcal{D}_n, \, \mathcal{S} \in \mathcal{E}(P_{\mathrm{obs}})) = \frac{\Pi(\mathrm{d}\mathcal{S})}{\Pi(\mathcal{E}(P_{\mathrm{obs}}))}. \tag{17}$$

Combining with Step 2 yields

$$T_n \longrightarrow \frac{1}{\Pi(\mathcal{E}(P_{\mathrm{obs}}))} \int_{\mathcal{E}(P_{\mathrm{obs}})} Q(\mathbb{P}_{\mathrm{int}}^{\mathcal{S}}) \, \Pi(\mathrm{d}\mathcal{S}), \qquad n \to \infty. \tag{18}$$

**Step 4: The limit cannot equal the identifiable target.**  Split the integral over $\mathcal{E}(P_{\mathrm{obs}})$ into $\mathcal{W}$ and $\mathcal{Z} \cap \mathcal{E}(P_{\mathrm{obs}})$:

$$\int_{\mathcal{E}(P_{\mathrm{obs}})} Q(\mathbb{P}_{\mathrm{int}}^{\mathcal{S}}) \, \Pi(\mathrm{d}\mathcal{S}) = \int_{\mathcal{W}} Q(\mathbb{P}_{\mathrm{int}}^{\mathcal{S}}) \, \Pi(\mathrm{d}\mathcal{S}) + \int_{\mathcal{Z} \cap \mathcal{E}(P_{\mathrm{obs}})} Q(\mathbb{P}_{\mathrm{int}}^{\mathcal{S}}) \, \Pi(\mathrm{d}\mathcal{S}). \tag{19}$$

By definition of $\mathcal{W}$ and identifiability, for every $\mathcal{S} \in \mathcal{W}$ we have $Q(\mathbb{P}_{\mathrm{int}}^{\mathcal{S}}) = \bar{Q}(P_{\mathrm{obs}})$, and thus

$$\int_{\mathcal{W}} Q(\mathbb{P}_{\mathrm{int}}^{\mathcal{S}}) \, \Pi(\mathrm{d}\mathcal{S}) = \bar{Q}(P_{\mathrm{obs}}) \, \Pi(\mathcal{W}). \tag{20}$$

Hence,

$$\lim_{n \to \infty} T_n = \frac{\bar{Q}(P_{\mathrm{obs}}) \, \Pi(\mathcal{W}) + \int_{\mathcal{Z} \cap \mathcal{E}(P_{\mathrm{obs}})} Q(\mathbb{P}_{\mathrm{int}}^{\mathcal{S}}) \, \Pi(\mathrm{d}\mathcal{S})}{\Pi(\mathcal{W}) + \Pi\big(\mathcal{Z} \cap \mathcal{E}(P_{\mathrm{obs}})\big)}. \tag{21}$$

By the non-cancellation assumption (applied to this $P_{\mathrm{obs}}$) and $w > 0$, this limit is not equal to $\bar{Q}(P_{\mathrm{obs}})$.

Finally, since $\mathcal{D}_n \sim \mathbb{P}_{\mathrm{obs}}^{\mathcal{S}^w}$ and $\mathcal{C}$ is identifiable,

$$Q(\mathbb{P}_{\mathrm{int}}^{\mathcal{S}^w}) = \bar{Q}(P_{\mathrm{obs}}). \tag{22}$$

Thus $\lim_{n \to \infty} T_n \neq Q(\mathbb{P}_{\mathrm{int}}^{\mathcal{S}^w})$, contradicting that $\Pi$ is well-specified for $\mathcal{C}$. Therefore, it must be that $\Pi(\mathcal{Z}) = 0$.  $\square$

## C.2 PROOF OF LEMMA B.1 (LINEAR IV)

*Proof of Lemma B.1.* We prove that for any $\zeta \neq \zeta^*$, there exists an SCM $S = (\alpha, \beta, \gamma, \delta, \zeta, \eta, \theta, \kappa \neq 0)$ that induces the same observational distribution as $S^*$.

**Step 1: Observational distribution of $S^*$**

The observational distribution is characterized by the covariance matrix $\Sigma^*$ of $(Z^*, A^*, Y^*)$:

$$\text{Var}(Z^*) = (\alpha^*)^2 \tag{23}$$

$$\text{Cov}(Z^*, A^*) = (\alpha^*)^2 \beta^* \tag{24}$$

$$\text{Var}(A^*) = (\alpha^* \beta^*)^2 + (\delta^*)^2 + (\gamma^*)^2 \tag{25}$$

$$\text{Cov}(Z^*, Y^*) = \zeta^* (\alpha^*)^2 \beta^* \tag{26}$$

$$\text{Cov}(A^*, Y^*) = \zeta^* [(\alpha^* \beta^*)^2 + (\delta^*)^2 + (\gamma^*)^2] + \eta^* \gamma^* \tag{27}$$

$$\text{Var}(Y^*) = \zeta^{*2} [(\alpha^* \beta^*)^2 + (\delta^*)^2 + (\gamma^*)^2] + 2\zeta^* \eta^* \gamma^* + (\eta^*)^2 + (\theta^*)^2 \tag{28}$$

**Step 2: Construction of alternative SCM $S$**

The covariance matrix $\Sigma$ has elements:

$$\text{Var}(Z) = \alpha^2 + \kappa^2 \tag{29}$$

$$\text{Cov}(Z, A) = \alpha^2 \beta + \kappa(\kappa\beta + \gamma) \tag{30}$$

$$\text{Var}(A) = (\alpha\beta)^2 + (\kappa\beta + \gamma)^2 + \delta^2 \tag{31}$$

$$\text{Cov}(Z, Y) = \zeta(\alpha^2 \beta + \kappa(\kappa\beta + \gamma)) + \eta\kappa \tag{32}$$

$$\text{Cov}(A, Y) = \zeta[(\alpha\beta)^2 + (\kappa\beta + \gamma)^2 + \delta^2] + \eta(\kappa\beta + \gamma) \tag{33}$$

$$\text{Var}(Y) = \zeta^2[(\alpha\beta)^2 + (\kappa\beta + \gamma)^2 + \delta^2] + 2\zeta\eta(\kappa\beta + \gamma) + \eta^2 + \theta^2 \tag{34}$$

**Step 3: Parameter matching**

To achieve $\Sigma = \Sigma^*$, we need:

$$\alpha^2 + \kappa^2 = (\alpha^*)^2 \tag{35}$$

$$\alpha^2 \beta + \kappa(\kappa\beta + \gamma) = (\alpha^*)^2 \beta^* \tag{36}$$

$$(\alpha\beta)^2 + (\kappa\beta + \gamma)^2 + \delta^2 = (\alpha^* \beta^*)^2 + (\delta^*)^2 + (\gamma^*)^2 \tag{37}$$

$$\zeta(\alpha^2 \beta + \kappa(\kappa\beta + \gamma)) + \eta\kappa = \zeta^* (\alpha^*)^2 \beta^* \tag{38}$$

$$\zeta[(\alpha\beta)^2 + (\kappa\beta + \gamma)^2 + \delta^2] + \eta(\kappa\beta + \gamma) = \zeta^* [(\alpha^* \beta^*)^2 + (\delta^*)^2 + (\gamma^*)^2] + \eta^* \gamma^* \tag{39}$$

$$\zeta^2[(\alpha\beta)^2 + (\kappa\beta + \gamma)^2 + \delta^2] + 2\zeta\eta(\kappa\beta + \gamma) + \eta^2 + \theta^2 = (\text{Var}(Y^*)) \tag{40}$$

**Step 4: Solution construction**

We choose $\kappa \neq 0$ such that $|\kappa| < |\alpha^*|$. We set

$$\alpha = \sqrt{(\alpha^*)^2 - \kappa^2}, \tag{41}$$

$$\beta = \frac{(\alpha^*)^2 \beta^*}{\alpha^2 + \kappa^2} = \beta^* \quad \text{(from (35) and (36))}, \tag{42}$$

$$\delta = \delta^*, \tag{43}$$

$$\kappa\beta + \gamma = \pm\sqrt{(\gamma^*)^2 - (\alpha^* \beta^*)^2 + (\alpha\beta)^2} \quad \text{(from (37))}. \tag{44}$$

Since $\alpha\beta = \alpha\beta^* = \frac{\alpha}{\alpha^*}\alpha^*\beta^*$, we have .

$$(\alpha\beta)^2 = \frac{\alpha^2}{(\alpha^*)^2}(\alpha^*\beta^*)^2 = \frac{(\alpha^*)^2 - \kappa^2}{(\alpha^*)^2}(\alpha^*\beta^*)^2. \tag{45}$$

Therefore, we have

$$\kappa\beta + \gamma = \pm\sqrt{(\gamma^*)^2 + \frac{\kappa^2}{(\alpha^*)^2}(\alpha^*\beta^*)^2}. \tag{46}$$

From Eq. (38) and Eq. (39), we can solve for $\eta$ via

$$\eta = \frac{\zeta^*(\alpha^*)^2\beta^* - \zeta(\alpha^2\beta + \kappa(\kappa\beta + \gamma))}{\kappa}. \tag{47}$$

Finally, $\theta$ is determined from Eq. (40).

**Step 5: Existence Verification**

The system has 8 parameters $(\alpha, \beta, \gamma, \delta, \zeta, \eta, \theta, \kappa)$ and 6 constraints (the 6 unique entries of the covariance matrix). Since $\zeta \neq \zeta^*$ is fixed and $\kappa \neq 0$ is chosen, we have 6 remaining parameters for 6 constraints. The key observation is that the introduction of confounding ($\kappa \neq 0$) creates additional correlation structures that can compensate for the change in the causal effect $\zeta$, allowing the observational distribution to remain unchanged.

$\square$

*Proof of Lemma B.2.* Let $\mathcal{S} = (\alpha, \beta, \gamma, \delta, \zeta, \eta, \theta)$ be any SCM from the linear IV class in Eq. (10). Then, the following coefficients are identified via observational data alone:

$$\alpha = \sqrt{\mathrm{Var}(Z)}, \tag{48a}$$
$$\beta = \mathbb{E}[Y \mid Z = 1], \tag{48b}$$
$$\zeta = \frac{\zeta\beta}{\beta} = \frac{\mathbb{E}[\zeta(\beta + \delta\epsilon_A)] - \mathbb{E}[\zeta(\delta\epsilon_A)]}{\beta} = \frac{\mathbb{E}[Y \mid Z = 1] - \mathbb{E}[Y \mid Z = 0]}{\mathbb{E}[A|Z = 1] - \mathbb{E}[A|Z = 0]} \tag{48c}$$

as well as the combination of coefficients

$$\delta^2 + \gamma^2 = \mathrm{Var}(A \mid Z), \qquad \text{and} \qquad \eta^2 + \theta^2 = \mathrm{Var}(Y \mid A) \tag{49}$$

and the back-door adjustment

$$\mathbb{E}[Y \mid A = 1] - E[Y \mid A = 0] = \zeta + \eta(\mathbb{E}[U \mid A = 1] - \mathbb{E}[U \mid A = 0]) \tag{50}$$
$$= \zeta + \frac{\eta\gamma}{\gamma^2 + \delta^2 + (\beta\alpha)^2}. \tag{51}$$

Note that the back-door adjustment is biased for $\zeta$ due to unobserved confounding.

**Noiseless treatment case.** Let now $\mathcal{S}^* = (\alpha^*, \beta^*, \gamma^*, \delta^*, \zeta^*, \eta^*, \theta^*)$ denote an arbitrary fixed SCM from the linear IV class. We start by constructing $\mathcal{S}_1 = (\alpha_1, \beta_1, \gamma_1, \delta_1 = 0, \zeta_1, \eta_1, \theta_1)$ such that $\mathbb{P}_{\mathrm{obs}}^{\mathcal{S}_1} = \mathbb{P}_{\mathrm{obs}}^{\mathcal{S}^*}$. Because of Eq. (48), we set

$$\alpha_1 = \alpha^*, \; \beta_1 = \beta^*, \; \zeta_1 = \zeta^*. \tag{52}$$

Furthermore, setting $\delta_1 = 0$ implies due to Eq. (49) that

$$\gamma_1^2 = \delta^{*2} + \gamma^{*2}. \tag{53}$$

Due to Eq. (50), it must holds that

$$\frac{\eta_1\gamma_1}{\gamma_1^2 + (\beta_1\alpha_1)^2} = \frac{\eta^*\gamma^*}{\delta^{*2} + \gamma^{*2} + (\beta^*\alpha^*)^2}, \tag{54}$$

which implies that

$$\eta_1^2 = \frac{\eta^{*2}\gamma^{*2}}{\delta^{*2} + \gamma^{*2}}. \tag{55}$$

Finally, due to Eq. (49), we yield

$$\theta_1^2 = \eta^{*2} + \theta^{*2} - \frac{\eta^{*2}\gamma^{*2}}{\delta^{*2} + \gamma^{*2}}, \tag{56}$$

which means that every parameter of $\mathcal{S}_1$ has a unique solution in terms of parameters of $\mathcal{S}^*$ under the constraints of preserving the observational distribution.

**Noiseless outcome case.** We now construct $\mathcal{S}_2 = (\alpha_2, \beta_2, \gamma_2, \delta_2, \zeta_2, \eta_2, \theta_2 = 0)$ such that $\mathbb{P}^{\mathcal{S}_2}_{\mathrm{obs}} = \mathbb{P}^{\mathcal{S}^*}_{\mathrm{obs}}$. Again, Eq. (48) implies that

$$\alpha_2 = \alpha^*, \ \beta_2 = \beta^*, \ \zeta_2 = \zeta^*, \tag{57}$$

and setting $\theta_2 = 0$ implies due to Eq. (49) that

$$\eta_2^2 = \eta^{*2} + \theta^{*2}. \tag{58}$$

Due to Eq. (50), it must holds that $\frac{\eta\gamma}{\gamma^2 + \delta^2 + (\beta\alpha)^2} = \frac{\eta\gamma}{\delta^{*2} + \gamma^{*2} + (\beta^*\alpha^*)^2}$ which implies that

$$\gamma_2^2 = \frac{\eta^{*2}\gamma^{*2}}{\eta^{*2} + \theta^{*2}}. \tag{59}$$

Finally, due to Eq. (49), we have

$$\delta^2 = \eta^{*2} + \gamma^{*2} - \frac{\eta^{*2}\gamma^{*2}}{\eta^{*2} + \theta^{*2}}, \tag{60}$$

which means that every parameter of $\mathcal{S}_2$ has a unique solution in terms of parameters of $\mathcal{S}^*$ under the constraints of preserving the observational distribution. $\qquad\square$

# D  IMPLEMENTATION DETAILS

## D.1  IMPLEMENTATION DETAILS OF DATA PRIOR

**Data Prior.** (i) For each covariate cluster $C_i$ containing latent nodes, we sample a random MLP-style graph over $\text{pa}(C_i)$ by drawing biases and edge-weights from $\Pi_{C_i}$ and then pruning edges at random to ensure acyclicity. We evaluate this graph with tanh activations and noise (from normal, uniform, Laplace, or logistic distribution) to produce continuous features, then apply randomized thresholds to discretize or binarize a subset, yielding mixed-type covariates via our unstructured BNN prior. (ii) For treatment (and outcome) clusters $C_j$ of purely observed nodes, we instantiate a second BNN $f_\theta^{(j)}$ over $\text{pa}(C_j)$ (with $\theta \sim \Pi_{C_j}$ and the same acyclicity constraint). We forward-propagate the covariates through $f_\theta^{(j)}$ with injected noise to compute a scalar propensity score, then threshold to assign a binary treatment. We forward-propagate both covariates and treatment to obtain potential outcomes. The resulting treatment (and outcome) are sampled from our structured BNN prior.

We sample covariates from a DAG-structured SCM by drawing a random MLP-like directed graph and assigning each node a bias, edge weights sampled from prior distributions. The resulting MLP-like graph is transformed into a DAG by randomly dropping edges, and structural equations with tanh activations and heterogeneous noise distributions (normal, uniform, Laplace, or logistic) generate continuous features. Then we apply a randomized feature transformation that discretizes some features and binarizes others, yielding mixed-type covariates. Next, we assign binary treatments via a separate randomly instantiated MLP and forward-propagate each covariate with injected noise to compute a propensity score.

**Input format.** CausalFM operates as an in-context learner like other foundation models, meaning it approximates Bayesian inference by conditioning on a dataset provided in its context window. Therefore, it requires an entire dataset to make predictions for specific query samples. (1) Input structure: The model accepts a dataset $\mathcal{D}_n = \{(x_i, a_i, y_i)\}_{i=1}^n$ acting as the context (or support set) and a query point $x_{\text{query}}$ (or a batch of query points). (2) Mechanism: The transformer processes the entire sequence of observed data $\mathcal{D}_n$ using self-attention to extract context-dependent representations. It then outputs the posterior predictive distribution for the causal quantity (e.g., CATE given the context $\mathcal{D}_n$ and the specific query $x_{\text{query}}$. (3) Comparison to fine-tuning baselines: In practice, standard baselines require an explicit training phase on a training set before evaluation. In contrast, CausalFM takes the "training" data as the input context (support set) and directly generates predictions for the test data (query set) in a single forward pass.

## D.2  IMPLEMENTATION DETAILS OF OUR METHOD

We encode observational data as tokens, and the embedded tokens are then processed through a transformer where attention is applied between the observations. We use transformer-based PFN as an encoder to extract a task- or context-dependent representation from input data. This representation is then passed to a Gaussian mixture model (GMM) head, which predicts the parameters of a GMM, including mixture weights, means, and standard deviations. The model outputs a mixture distribution over the target variable, and is trained end-to-end using the negative log-likelihood (NLL) of the observed targets under the predicted GMM. This enables uncertainty-aware and multi-modal predictions while leveraging the few-shot generalization capabilities of our model.

We instantiate a per-feature transformer tailored to CATE estimation. For a mini-batch with sequence length $S = S_{\text{supp}} + S_{\text{query}}$ (query set followed by support set). Confounders $X \in \mathbb{R}^{S \times B \times F_x}$, treatment $A \in \mathbb{R}^{S \times B \times F_a}$, and factual outcomes $Y \in \mathbb{R}^{S \times B \times F_y}$ are encoded as tokens. To prevent label leakage, we split at $S_{\text{supp}} = \lfloor 0.8\,S \rfloor$ and set $A_t = \text{NaN}$ and $Y_t = \text{NaN}$ for $t \geq S_{\text{supp}}$ (on the query set). The model thus observes $(X, A, Y)$ on support steps and learn to infer CATEs for the query set from $X$ only.

The $X$ stream uses a feature encoder, while $A$ and $Y$ pass through a NaN-indicator handler followed by feature projections. We concatenate the three streams along the token axis to obtain $H_0 \in \mathbb{R}^{B \times S \times (F_g+2) \times E}$, add a feature-token positional embedding, and process $H_0$ with $L$ transformer encoder blocks (self-attention only). We pool over tokens to produce feature $Z \in \mathbb{R}^{B \times S \times E}$. After lightweight MLP maps $Z$ to a scalar, a 1D $K$-component GMM head outputs mixture parameters $(\pi, \mu, \sigma)$ via $\pi = \text{softmax}(W_\pi z/T), \mu = W_\mu z, \sigma = \text{softplus}(W_\sigma z) + \varepsilon$, for each $z \in \mathbb{R}^E$. Our

training loss is the Gaussian-mixture negative log-likelihood (GMM-NLL). We thus obtain the distribution

$$p(\tau \mid x) = \sum_{k=1}^{K} \pi_k(x)\mathcal{N}\left(\mu_k(x), \sigma_k^2(x)\right) \tag{61}$$

And the CATE can be computed through

$$\hat{\tau}(x) = \mathbb{E}[\tau \mid x] = \sum_{k=1}^{K} \pi_k(x)\mu_k(x) \tag{62}$$

We use embedding size $E = 128$, $n_{\text{heads}} = 4$, feed-forward dimension $4E$, $L = 10$ encoder layers, GELU activations, and feature grouping size $= 1$ (per-feature tokens). For the GMM head we set $K = 5$, temperature $T = 1.0$, and variance floor $\varepsilon = 10^{-3}$. We train with Adam (learning rate $10^{-3}$, weight decay $10^{-5}$), batch size 16, and up to 150 epochs. We use early stopping on validation loss. Empirically, the total training time for causalFM model is about 24 hours on an NVIDIA A100 GPU.

We implement our CausalFM using PyTorch. Our model implementation builds upon the TabPFN architecture (Hollmann et al., 2023) from `https://github.com/PriorLabs/TabPFN/tree/main`.

### D.3 IMPLEMENTATION DETAILS OF BASELINES

For the standard CATE setting baselines, we follow the implementation from `https://github.com/AliciaCurth/CATENets/tree/main` for most of the CATE estimators, including S-learner (Künzel et al., 2019), T-learner (Künzel et al., 2019), TARNet (Shalit et al., 2017b), X-leaner (Künzel et al., 2019), DR-learner (Kennedy, 2023b), RA-learner (Curth & van der Schaar, 2021). For the foundation model baselines, we follow the author implementation from `https://github.com/vdblm/CausalPFN/tree/main` for CausalPFN (Balazadeh et al., 2025); we follow the author implementation from `https://github.com/jr2021/Do-PFN` for DoPFN (Robertson et al., 2025).

For the IV setting, we follow the implementation from `https://github.com/DennisFrauen/MRIV-Net/tree/main/models` for the most of the IV methods, including KIV (Singh et al., 2019), DFIV (Xu et al., 2021), DeepIV (Hartford et al., 2017), DeepGMM (Bennett et al., 2019), DMLIV (Syrgkanis et al., 2019). For each dataset and method, we evaluated 5 repetitions, each with a different random seed. All methods used the same train-test split.

# E  SYNTHETIC DATA GENERATION FOR THE STANDARD CATE ESTIMATION SETTING

We construct the standard CATE estimation datasets by sampling covariates $X$, treatment $A$, and continuous outcomes $Y$. The design induces rich nonlinearity while preserving strong ignorability $(A \perp\!\!\!\perp \{Y(0), Y(1)\} \mid X)$.

## E.1  COVARIATES VIA A DAG-STRUCTURED SCM

We first sample a layered directed graph (an MLP-like DAG), then evaluate a structural causal model (SCM) on its nodes and expose a random subset as observed features.

**Graph.**   Sample number of layers $L_X$ and hidden size $H_X$ from simple discrete priors (see "Hyper-parameters" below). Build a layered graph with $H_X$ nodes per layer and fully connect layer $\ell$ to $\ell+1$. Randomly drop a fraction $p_{\mathrm{drop}}^X$ of inter-layer edges to sparsify while keeping acyclicity.

**Node equations and noise.**   For each node $j$, sample weights $\{w_{jk}\}_{k \in \mathrm{pa}(j)}$, bias $b_j$, and an exogenous noise distribution $\varepsilon_j \sim \mathcal{D}_j$, where $\mathcal{D}_j$ is drawn from a meta-prior over {Normal, Uniform, Laplace, Logistic} with a random scale. Nodes are evaluated in topological order:

$$s_j = \sum_{k \in \mathrm{pa}(j)} w_{jk}\, x_k + b_j + \varepsilon_j, \qquad x_j = \tanh(s_j), \tag{63}$$

with the convention $\sum_{k \in \varnothing}(\cdot) = 0$ for roots. Let $U_X = \{\varepsilon_j\}$ denote the collection of all node noises.

**Observed features.**   Sample a feature index set $\mathcal{F} \subseteq V$ with $|\mathcal{F}| = d$ uniformly from all graph nodes. A single observation $X \in \mathbb{R}^d$ is obtained by re-sampling $U_X$, evaluating (63) over the DAG, and reading out $X = (x_j)_{j \in \mathcal{F}}$. Each sample uses independent $U_X$.

**Feature typing and transformations (Optional).**   Each selected feature $x_j$ is assigned a random type from {continuous, binary, categorical}. Continuous features are kept in their raw form $x_j \in (-1, 1)$. Binary features are obtained by mapping $x_j$ through a logistic function and drawing a Bernoulli sample. For categorical features, we first sample a base distribution $\pi^0 \in \Delta^{K-1}$ over $K$ categories from a Dirichlet prior. To make the distribution depend on the DAG value $x_j$, we introduce a fixed direction vector $v \in \mathbb{R}^K$ (normalized) and scale $\alpha > 0$, and form

$$\pi(x_j) = \mathrm{softmax}(\log \pi^0 + \alpha\, x_j\, v). \tag{64}$$

The observed categorical feature is then sampled as $X_i \sim \mathrm{Categorical}(\pi(x_j))$.

## E.2  TREATMENT ASSIGNMENT

Given $X$, we compute a stochastic logit via a feed-forward network with layer-wise exogenous noise and then sample a Bernoulli treatment $A \sim f_A(X, U_A)$.

**Network.**   Sample depth $L_A \geq 3$ and hidden width $H_A$. Let $h^{(0)} = X \in \mathbb{R}^d$ be the input layer. For hidden layers $\ell = 1, \dots, L_A - 1$,

$$s^{(\ell)} = W^{(\ell)} h^{(\ell-1)} + b^{(\ell)} + \varepsilon^{(\ell)}, \qquad h^{(\ell)} = \tanh(s^{(\ell)}), \tag{65}$$

and the (scalar) output logit

$$s_A = w^\top h^{(L_A - 1)} + b + \varepsilon^{(L_A)}. \tag{66}$$

We define the propensity $p = \sigma(s_A)$ and sample

$$A \sim \mathrm{Bernoulli}(p). \tag{67}$$

Let $U_A = \left(\varepsilon^{(\ell)}{}_{\ell=1}^{L_A}, , U_B\right)$ collect all exogenous noises of the network and the random variable $U_B$ used for the Bernoulli sampling.

### E.3 CONTINUOUS OUTCOME

For each unit, we compute the potential outcomes $Y(0)$ and $Y(1)$ using the *same* exogenous noise $U_Y$.

**Network.** Sample depth $L_Y \geq 3$ and width $H_Y$; optionally drop a fraction $p_{\text{drop}}^Y$ of hidden edges to induce sparsity. For a given treatment level $a \in \{0, 1\}$, the input is $X$ and $A$, then for hidden layers

$$t^{(\ell)}(a) = V^{(\ell)} h^{(\ell-1)}(a) + c^{(\ell)} + \xi^{(\ell)}, \qquad h^{(\ell)}(a) = \tanh\big(t^{(\ell)}(a)\big), \qquad (68)$$

with $h^{(0)}(a) = [X, a]$, and the scalar output logit

$$Y(a) = v^\top h^{(L_Y-1)}(a) + c + \xi^{(L_Y)}. \qquad (69)$$

The factual outcome is

$$Y = A Y(1) + (1 - A) Y(0) \qquad (70)$$

Let $U_Y = \{\xi^{(\ell)}\}_{\ell=1}^{L_Y}$ denote outcome-network noises; *the same $U_Y$ is reused when constructing $Y(0)$ and $Y(1)$ for the same unit.*

### E.4 INDEPENDENCE AND IDENTIFICATION

All exogenous noises are sampled independently across mechanisms and samples: $U_X \perp U_A \perp U_Y$ and i.i.d. across units. Hence strong ignorability holds:

$$A \perp\!\!\!\perp \{Y(0), Y(1)\} \mid X, \qquad 0 < \Pr(A{=}1 \mid X) < 1, \qquad (71)$$

with overlap ensured by the sigmoid in (66) to (67).

### E.5 HYPERPARAMETERS AND PRIORS (AS USED IN OUR CODE)

We use simple, reproducible priors for architecture, weights, and noises:

- **Covariate DAG:** $L_X \sim \text{Unif}\{3, 4, 5, 6\}$, $H_X \sim \text{Unif}\{15, \ldots, 40\}$, edge-drop $p_{\text{drop}}^X{=}0.5$.
- **Treatment net:** $L_A \sim \text{Unif}\{3, 4\}$, $H_A \sim \text{Unif}\{8, \ldots, 20\}$.
- **Outcome net:** $L_Y \sim \text{Unif}\{3, 4, 5\}$, $H_Y \sim \text{Unif}\{10, \ldots, 25\}$, edge-drop $p_{\text{drop}}^Y{=}0.4$.
- **Weights/biases:** i.i.d. $w, b \sim \mathcal{N}(0, \sigma_w^2)$ with task-specific $\sigma_w$.
- **Node noises:** for each node, draw a type in {Normal, Uniform, Laplace, Logistic} and a scale from a wide range; sample fresh noises per unit and layer as in (63), (65)–(66), (68)–(69).
- **Activation:** $\tanh$ for all hidden layers; output layers are linear (logits).
- **Features observed:** choose $\mathcal{F}$ uniformly at random from all DAG nodes, $|\mathcal{F}| = d$.

### E.6 GENERATION PIPELINE

For each dataset:

1. Sample the covariate DAG, parameters, and noises; for each unit, evaluate the DAG in topological order to obtain $X$ by reading nodes in $\mathcal{F}$.
2. Given $X$, construct the treatment network with $U_A$ to get $p$ and sample $A \sim \text{Bernoulli}(p)$.
3. For outcomes, sample $U_Y$ once per unit and use it to compute $Y(0)$ and $Y(1)$ via (69).

### E.7 SYNTHETIC DATASETS SIZE

We sample 10000 synthetic training datasets from data prior with different data generation mechanism. Each training datasets contain 1024 data samples. The feature dimensions are also different across the datasets, ranging from 10 to 100. The features are mixed data type with continuous, binary and categorical.

# F   SYNTHETIC DATA GENERATION FOR THE INSTRUMENTAL VARIABLES (IV) SETTING

We aim at estimating CATEs from observational data under unobserved confounding using IVs. In contrast to the standard CATE setting, where strong unconfoundedness holds, our IV datasets intentionally violate unconfoundedness by introducing an unobserved confounder $U$ that affects both treatment $A$ and outcome $Y$. Identification is instead driven by an instrument $Z$ that (i) is relevant for $A$, (ii) has no direct path to $Y$ beyond $A$ (exclusion), and (iii) is conditionally independent of $U$ given $X$.

**Key differences vs. standard CATE.**   (i) *Ignorability is broken*: $A \not\perp\!\!\!\perp \{Y(0), Y(1)\} \mid X$ due to $U \to A$ and $U \to Y$. (ii) We introduce an *instrument $Z$* with $Z \perp U \mid X$, $Z \not\perp\!\!\!\perp A \mid X$, and no $Z \to Y$ edge (exclusion). (iii) Outcomes are generated via an *additive* structural form $Y = f(X, A) + g(X, U) + \varepsilon_Y$ with $f$ and $g$ deterministic neural networks; the same $\varepsilon_Y$ is reused across $Y(0)$ and $Y(1)$ for a unit to ensure counterfactual consistency.

## F.1   COVARIATES AND LATENT CONFOUNDERS VIA A DAG-STRUCTURED SCM

We reuse the DAG-SCM from the standard setting to produce a wide set of base variables $W$, then split it into observed covariates $X$ and unobserved confounders $U$. Thus we have different strength of the unobserved confounders from weak to sufficiently strong.

**Graph and node equations.**   Sample number of layers $L_X$ and hidden size $H_X$, build a layered DAG (fully connect layer $\ell$ to $\ell+1$), and drop a fraction $p_{\text{drop}}^X$ of inter-layer edges to sparsify. For each node $j$, sample weights $\{w_{jk}\}_{k \in \text{pa}(j)}$, bias $b_j$, and a node-specific exogenous noise $\varepsilon_j \sim \mathcal{D}_j$ (type and scale drawn once per node). Evaluate in topological order

$$s_j = \sum_{k \in \text{pa}(j)} w_{jk} \, v_k + b_j + \varepsilon_j, \qquad v_j = \tanh(s_j). \tag{72}$$

Draw a feature index set for $W = (v_j)$ with $|W| = d_X + d_U^{\max}$, and then sample the actual confounder dimension $d_U \in \{2, \ldots, 5\}$ uniformly. Split $U \in \mathbb{R}^{d_U}$ from the first $d_U$ coordinates of $W$, $X \in \mathbb{R}^{d_X}$ from the next $d_X$ coordinates. Node noises $\{\varepsilon_j\}$ are drawn independently per unit.

## F.2   INSTRUMENT VARIABLE

We generate $Z$ from $X$ only, ensuring $Z \perp U \mid X$ by construction and precluding any direct $U \to Z$ path. Let $\phi_Z$ be a feed-forward network with input $X$ and no layer-wise exogenous noise; the network parameters are sampled once per dataset and then fixed. For a unit with covariates $X$,

$$s_Z = \phi_Z(X), \qquad Z = \begin{cases} \text{Bernoulli}\big(\sigma(s_Z)\big), & \text{binary instrument}, \\ s_Z, & \text{continuous instrument}. \end{cases} \tag{73}$$

We randomly choose between the binary and continuous variants when creating datasets. Relevance is induced via the $Z \to A$ path in the treatment mechanism below.

Note that the instrument variable $Z$ has a direct influence on the treatment $A$, but does not have a direct effect on the outcome $Y$.

## F.3   TREATMENT VARIABLE

Given $(X, Z, U)$, treatment is generated via a deterministic network $\phi_A$ followed by a Bernoulli draw. There is *no* layer-wise noise inside $\phi_A$; the only randomness is the terminal Bernoulli. For a unit,

$$s_A = \phi_A\big([X; Z; U]\big), \qquad p = \sigma(s_A), \qquad A \sim \text{Bernoulli}(p). \tag{74}$$

This introduces $U \to A$ and hence breaks ignorability, while maintaining $Z \perp U \mid X$ and $Z \to A$ relevance.

### F.4 OUTCOME VARIABLES

The instrument variable $Z$ has no direct effect on the outcomes. Outcomes are generated additively from a treatment channel $f$ and a confounding channel $g$, both deterministic MLPs with inputs $[X; A]$ and $[X; U]$, respectively. Let $\varepsilon_Y \sim \mathcal{N}(0, \sigma_Y^2)$ be an i.i.d. scalar noise drawn once per unit,

$$Y(a) = f(X, a) + g(X, U) + \varepsilon_Y, \qquad a \in \{0, 1\}, \tag{75}$$

$$Y = A\,Y(1) + (1 - A)\,Y(0). \tag{76}$$

By construction there is no $Z \to Y$ edge (exclusion), since $Z$ influences $Y$ only through $A$.

### F.5 INDEPENDENCE AND IDENTIFICATION (IV)

All exogenous noises are sampled independently across units and mechanisms. The IV conditions hold by construction,

$$\text{(Independence)} \quad Z \perp U \mid X, \tag{77}$$

$$\text{(Exclusion)} \quad Y(a) \text{ depends on } X, a, U \text{ and } \varepsilon_Y \text{ only (no } Z), \tag{78}$$

$$\text{(Relevance)} \quad Z \not\perp A \mid X. \tag{79}$$

### F.6 HYPERPARAMETERS AND PRIORS

We use simple priors mirroring our implementation:

- **DAG-SCM for** $(X, U)$: $L_X \sim \text{Unif}\{2, 3, 4, 5\}$, $H_X \sim \text{Unif}\{10, \ldots, 50\}$, edge-drop $p_{\text{drop}}^X = 0.4$; node noises $\varepsilon_j$ draw a type in {Normal, Uniform, Laplace, Logistic} with random scale.
- **Instrument net** $\phi_Z$: depth $L_Z \geq 3$, width $H_Z \sim \text{Unif}\{8, \ldots, 30\}$; output is either Bernoulli with $\sigma(s_Z)$ (binary $Z$) or real-valued $s_Z$ (continuous $Z$); no layer-wise noise.
- **Treatment net** $\phi_A$: depth $L_A \geq 3$, width $H_A \sim \text{Unif}\{8, \ldots, 30\}$; *no* layer-wise noise; $A \sim \text{Bernoulli}(\sigma(s_A))$.
- **Outcome nets** $f, g$: depths $L_f, L_g \sim \text{Unif}\{3, \ldots, 6\}$, widths $H_f, H_g \sim \text{Unif}\{10, \ldots, 25\}$; $\varepsilon_Y \sim \mathcal{N}(0, \sigma_Y^2)$ with $\sigma_Y = 0.5$ by default.
- **Weights/biases:** i.i.d. $w, b \sim \mathcal{N}(0, 1)$ sampled once per dataset; $\tanh$ activations.
- **Strength:** $d_U \sim \text{Unif}\{2, \ldots, 5\}$.

### F.7 GENERATION PIPELINE (IV)

For each dataset, we execute:

1. Sample the covariate DAG and parameters; for each unit, evaluate (72) to obtain a wide matrix then split it into $(U, X)$.
2. Given $X$, compute the instrument $Z$ via (73) (binary or continuous and mixed).
3. Given $(X, Z, U)$, compute the treatment propensity $p = \sigma(s_A)$ via (74) and sample $A \sim \text{Bernoulli}(p)$.
4. Draw a single $\varepsilon_Y$ per unit and compute $Y(0), Y(1)$ using (75); set the factual outcome by (76).

This yields datasets matching the classical IV graph and enabling evaluation of IV estimators.

# G  SYNTHETIC DATA GENERATION FOR THE FRONT-DOOR–ADJUSTED SETTING

## G.1  FRONT-DOOR ADJUSTMENT DATASETS

We next construct datasets satisfying the *front-door* criterion. Besides covariates $X$, treatment $A$, and continuous outcomes $Y$, we introduce a mediator $M$. The design ensures that $A$ affects $Y$ only through $M$ (no direct $A \to Y$ path), $U$ (unobserved) confounds $A$ and $Y$ but does not affect $M$.

**Covariates via a DAG-structured SCM.**  Identical to the standard setting: we sample a layered DAG, draw node-wise weights/biases/noise, evaluate in topological order as in (63), and expose $d$ node values as observed features $X \in \mathbb{R}^d$. Independent exogenous noises $U_X = \{\varepsilon_j\}$ are re-sampled per unit.

**Latent confounders.**  From the same SCM evaluation we also retain $q$ additional node values as unobserved confounders $U \in \mathbb{R}^q$ (not revealed to learners). These induce confounding between $A$ and $Y$.

### G.1.1  TREATMENT ASSIGNMENT WITH LATENT CONFOUNDING

Given $(X, U)$, we sample a feed-forward network and generate treatment. Let $L_A \geq 3$ and $H_A$ be the depth and width, respectively. With $h^{(0)} = [X, U]$,

$$s_A^{(\ell)} = W_A^{(\ell)} h^{(\ell-1)} + b_A^{(\ell)}, \quad h^{(\ell)} = \tanh(s_A^{(\ell)}), \ \ \ell = 1, \dots, L_A - 1, \tag{80}$$

and scalar logit

$$\tilde{s}_A = w_A^\top h^{(L_A - 1)} + b_A, \qquad p = \sigma(\tilde{s}_A), \qquad A \sim \text{Bernoulli}(p). \tag{81}$$

### G.1.2  MEDIATOR MECHANISM

The mediator is generated from $(X, A)$ only, thereby enforcing the front-door exclusion $U \nrightarrow M$. Let $L_M \geq 3$, $H_M$ be depth and width, with input $g^{(0)} = [X, A]$,

$$r^{(\ell)} = W_M^{(\ell)} g^{(\ell-1)} + b_M^{(\ell)} + \varepsilon_M^{(\ell)}, \qquad g^{(\ell)} = \tanh(r^{(\ell)}), \ \ \ell = 1, \dots, L_M - 1, \tag{82}$$

and scalar output

$$M = w_M^\top g^{(L_M - 1)} + b_M + \varepsilon_M^{(L_M)}. \tag{83}$$

We denote $U_M = \{\varepsilon_M^{(\ell)}\}_{\ell=1}^{L_M}$.

### G.1.3  OUTCOME VARIABLE

Outcomes are constructed to satisfy $A \to M \to Y$ as the *only* causal path from $A$ to $Y$, while allowing $U \to Y$ and $X \to Y$. We decompose $Y$ into an $M$-path component and a confounding component:

$$\textit{Mediator path:} \quad r_Y^{(\ell)} = V^{(\ell)}[h^{(\ell-1)}] + c^{(\ell)} + \xi^{(\ell)}, \ \ h^{(0)} = [X, M], \ \ h^{(\ell)} = \tanh(r_Y^{(\ell)}),$$
$$R(X, M) = v^\top h^{(L_Y - 1)} + c + \xi^{(L_Y)}, \tag{84}$$

$$\textit{Confounding path:} \quad G(X, U) = \tilde{v}^\top \tilde{h}^{(L_G - 1)} + \tilde{c} + \tilde{\xi}^{(L_G)}, \ \ \tilde{h}^{(0)} = [X, U], \ \ \tilde{h}^{(\ell)} = \tanh(\cdot), \tag{85}$$

and define the potential outcomes

$$Y(a) = R(X, M(a)) + G(X, U) + \epsilon_Y, \qquad M(a) \text{ computed from (82)–(83) with } A = a. \tag{86}$$

The factual outcome is $Y = A\, Y(1) + (1 - A)\, Y(0)$. By construction there is no direct $A \to Y$ edge; $A$ influences $Y$ solely via $M$.

### G.1.4 HYPERPARAMETERS AND PRIORS

- **Covariate DAG:** $L_X \sim \text{Unif}\{3, 4, 5, 6\}$, $H_X \sim \text{Unif}\{15, \ldots, 40\}$, edge-drop $p_{\text{drop}}^X{=}0.5$; node noises drawn per-node from {Normal, Uniform, Laplace, Logistic} with random scale.
- **Treatment net** (Eq. (80)–(81)): $L_A \sim \text{Unif}\{3, 4\}$, $H_A \sim \text{Unif}\{8, \ldots, 20\}$.
- **Mediator net** (Eq. (82)–(83)): $L_M \sim \text{Unif}\{3, 4\}$, $H_M \sim \text{Unif}\{8, \ldots, 20\}$.
- **Outcome nets** (Eq. (84)–(86)): $L_Y, L_G \sim \text{Unif}\{3, 4, 5\}$, widths $\sim \text{Unif}\{10, \ldots, 25\}$; additive Gaussian $\epsilon_Y$ with task-specific scale.
- **Weights/biases:** i.i.d. $\mathcal{N}(0, \sigma_w^2)$; $\tanh$ nonlinearity.

### G.1.5 GENERATION PIPELINE

For each dataset:

1. Sample the covariate DAG and evaluate to obtain $(X, U)$ (observed $X$, hidden $U$).
2. Compute $p(A{=}1 \mid X, U)$ via (80)–(81) and sample $A$.
3. Evaluate the mediator $M$ from $(X, A)$ using (82)–(83).
4. Sample $U_Y$ once per unit and compute $Y(0)$ and $Y(1)$ via (86) by first obtaining $M(0)$ and $M(1)$ from the mediator net; set $Y = A\,Y(1) + (1{-}A)\,Y(0)$.

# H ADDITIONAL EXPERIMENTS

## H.1 EVALUATION IN THE FRONT-DOOR ADJUSTMENT SETTING

### H.1.1 BASELINES FOR FRONT-DOOR ADJUSTMENT SETTING

In contrast to the standard CATE or IV settings, there are few established baselines for the front-door case. Identification in this setting is enabled through Pearl's front-door formula (Pearl, 2009). The natural baseline is therefore the **plug-in front-door learner**, which estimates the necessary nuisance components, i.e., $P(M \mid A, X)$, $P(A \mid X)$, and $\mathbb{E}[Y \mid M, X]$ and substitutes them into the identification formula to recover causal quantities. To assess the role of model flexibility in estimating these nuisance functions, we implement the plug-in learner with different regression methods, including linear regression, Random Forests, and neural networks.

### H.1.2 RESULTS FOR FRONT-DOOR ADJUSTMENT SETTING

Table 4 reports the averaged PEHE across datasets. We observe that CausalFM achieves competitive CATE estimation. Importantly, these results hold *without requiring model retraining* for our model, demonstrating the adaptability of our approach to the front-door setting.

## H.2 ADDITIONAL RESULTS ON THE STANDARD CATE ESTIMATION

We report the detailed standard CATE estimation on 10 synthetic datasets in Table 5. We show our method gives the best estimation on most of the datasets.

Table 5: Standard CATE estimation on 10 synthetic datasets.

| Method | $D_1$ | $D_2$ | $D_3$ | $D_4$ | $D_5$ | $D_6$ | $D_7$ | $D_8$ | $D_9$ | $D_{10}$ |
|---|---|---|---|---|---|---|---|---|---|---|
| BASELINES (A): STANDARD CATE ESTIMATORS | | | | | | | | | | |
| S-learner (Künzel et al., 2019) | 0.725 | 0.583 | 0.752 | 0.829 | 0.614 | 0.892 | 0.858 | 0.421 | 0.680 | 0.985 |
| T-learner (Künzel et al., 2019) | 0.652 | 0.496 | 0.666 | 0.746 | 0.552 | 0.849 | 0.761 | 0.357 | 0.608 | 0.931 |
| TARNet (Shalit et al., 2017b) | 0.769 | 0.779 | 0.817 | 0.984 | 0.640 | 0.938 | 1.405 | 0.505 | 0.736 | 0.968 |
| RA-learner (Curth & van der Schaar, 2021) | 0.620 | 0.421 | 0.644 | 0.706 | 0.523 | 0.808 | 0.646 | 0.353 | 0.613 | 0.759 |
| X-learner (Künzel et al., 2019) | 0.574 | 0.400 | 0.614 | 0.634 | 0.381 | 0.713 | 0.686 | 0.302 | 0.549 | 0.779 |
| DR-learner (Kennedy, 2023b) | 0.783 | 0.533 | 0.767 | 0.947 | 0.867 | 0.882 | 0.791 | 0.4230 | 0.653 | 0.998 |
| BASELINES (B): FOUNDATION MODELS-BASED METHODS | | | | | | | | | | |
| CausalPFN (Balazadeh et al., 2025) | 0.493 | 0.489 | 0.585 | 0.743 | 0.413 | 0.615 | 0.950 | 0.288 | 0.453 | 0.544 |
| DoPFN (Robertson et al., 2025) | 0.417 | 0.313 | 0.228 | 0.679 | 0.591 | 0.475 | 0.497 | 0.551 | 0.610 | 0.827 |
| **CausalFM (ours)** | 0.454 | 0.487 | 0.515 | 0.677 | 0.204 | 0.618 | 0.950 | 0.278 | 0.442 | 0.532 |

Reported: PEHE (Lower = better, best in bold).

## H.3 RESULTS ON OTHER DATASETS

In the following, we present detailed results of the experiments with ACIC 2016 datasets. We follow CausalPFN Balazadeh et al. (2025) obtaining data from `https://github.com/BiomedSciAI/causallib/tree/master/causallib/datasets/data/acic_challenge_2016` to evaluate on 10 different datasets with various data generation mechanism. The treatment and outcome were simulated from real-world data corresponding to 4802 individuals and 58 covariates. Table 6 shows the results of the CATE estimation.

Table 6: Standard CATE estimation on ACIC2016 datasets. Reported: PEHE (mean $\pm$ std.)

| Method | PEHE |
|---|---|
| BASELINES (A): STANDARD CATE ESTIMATORS | |
| S-learner (Künzel et al., 2019) | $1.191 \pm 0.15$ |
| T-learner (Künzel et al., 2019) | $1.143 \pm 0.14$ |
| TARNet (Shalit et al., 2017b) | $0.934 \pm 0.15$ |
| RA-learner (Curth & van der Schaar, 2021) | $0.762 \pm 0.14$ |
| X-learner (Künzel et al., 2019) | $0.519 \pm 0.16$ |
| DR-learner (Kennedy, 2023b) | $1.485 \pm 0.18$ |
| BASELINES (B): FOUNDATION MODELS-BASED METHODS | |
| CausalPFN (Balazadeh et al., 2025) | $0.239 \pm 0.11$ |
| DoPFN (Robertson et al., 2025) | $0.857 \pm 0.36$ |
| **CausalFM (ours)** | $0.638 \pm 0.32$ |

Lower = better (best in bold)

## H.4 ADDITIONAL RESULTS FOR THE IV SETTING

Table 7: IV setting for CATE estimation with binary instrument variable reported with PEHE. Results for benchmarking model performance across 10 different datasets under various confounding strength.

| Method | $D_1$ | $D_2$ | $D_3$ | $D_4$ | $D_5$ | $D_6$ | $D_7$ | $D_8$ | $D_9$ | $D_{10}$ |
|---|---|---|---|---|---|---|---|---|---|---|
| BASELINES (A): STANDARD CATE ESTIMATORS | | | | | | | | | | |
| TARNet (Shalit et al., 2017b) | 0.789 | 0.790 | 0.799 | 0.789 | 0.831 | 0.673 | 0.582 | 0.978 | 0.735 | 0.642 |
| DR-learner (Kennedy, 2023b) | 1.517 | 1.071 | 1.022 | 0.901 | 0.754 | 0.676 | 0.646 | 1.009 | 0.664 | 0.781 |
| BASELINES (B): STANDARD IV ESTIMATORS | | | | | | | | | | |
| KIV (Singh et al., 2019) | 0.660 | 0.344 | 0.340 | 0.394 | 0.544 | 0.460 | 0.299 | 0.731 | 0.532 | 0.241 |
| DFIV (Xu et al., 2021) | 0.654 | 0.245 | 1.022 | 0.459 | 1.145 | 0.770 | 0.741 | 0.366 | 0.971 | 0.717 |
| DeepIV (Hartford et al., 2017) | 0.614 | 0.300 | 0.310 | 0.372 | 0.514 | 0.404 | 0.309 | 0.706 | 0.510 | 0.235 |
| DeepGMM (Bennett et al., 2019) | 0.704 | 0.403 | 0.440 | 0.599 | 0.569 | 0.486 | 0.292 | 0.737 | 0.566 | 0.232 |
| DMLIV (Syrgkanis et al., 2019) | 0.712 | 0.379 | 0.361 | 0.433 | 0.548 | 0.450 | 0.293 | 0.722 | 0.549 | 0.344 |
| DRIV (Syrgkanis et al., 2019) | 0.869 | 0.470 | 0.353 | 0.368 | 0.565 | 0.448 | 0.272 | 0.715 | 0.587 | 0.667 |
| MRIV (Frauen & Feuerriegel, 2022) | 0.759 | 0.632 | 0.698 | 1.011 | 0.348 | 0.860 | 0.929 | 0.707 | 0.562 | 0.380 |
| BASELINES (C): FOUNDATION MODEL-BASED | | | | | | | | | | |
| DoPFN (Robertson et al., 2025) | 0.776 | 0.265 | 0.370 | 0.382 | 0.552 | 0.819 | 0.499 | 0.794 | 0.534 | 0.242 |
| CausalFM (ours) | 0.586 | 0.224 | 0.374 | 0.310 | 0.543 | 0.464 | 0.250 | 0.701 | 0.553 | 0.217 |

Reported: PEHE (mean ± standard deviation.) Lower = better (best in bold).

Table 8: IV setting for CATE estimation with continuous instrument variable reported with PEHE. Results for benchmarking model performance across 10 different datasets under various confounding strength.

| Method | $D_1$ | $D_2$ | $D_3$ | $D_4$ | $D_5$ | $D_6$ | $D_7$ | $D_8$ | $D_9$ | $D_{10}$ |
|---|---|---|---|---|---|---|---|---|---|---|
| BASELINES (A): STANDARD CATE ESTIMATORS | | | | | | | | | | |
| TARNet (Shalit et al., 2017b) | 0.943 | 0.825 | 1.025 | 0.458 | 1.007 | 1.316 | 0.848 | 1.004 | 0.825 | 0.884 |
| DR-learner (Kennedy, 2023b) | 1.038 | 1.055 | 0.946 | 0.533 | 0.955 | 1.071 | 1.109 | 1.502 | 1.258 | 0.888 |
| BASELINES (B): STANDARD IV ESTIMATORS | | | | | | | | | | |
| KIV (Singh et al., 2019) | 0.509 | 0.567 | 0.699 | 0.178 | 0.533 | 0.948 | 0.420 | 0.811 | 0.602 | 0.506 |
| DFIV (Xu et al., 2021) | 0.526 | 0.574 | 0.691 | 0.171 | 0.532 | 0.991 | 0.428 | 0.800 | 0.609 | 0.506 |
| DeepIV (Hartford et al., 2017) | 0.484 | 0.539 | 0.664 | 0.169 | 0.506 | 0.901 | 0.399 | 0.770 | 0.572 | 0.481 |
| DeepGMM (Bennett et al., 2019) | 0.543 | 0.581 | 0.682 | 0.165 | 0.532 | 1.035 | 0.437 | 0.789 | 0.615 | 0.505 |
| DMLIV (Syrgkanis et al., 2019) | 0.518 | 0.642 | 0.701 | 0.181 | 0.600 | 1.009 | 0.574 | 0.813 | 0.611 | 0.537 |
| DRIV (Syrgkanis et al., 2019) | 0.633 | 0.705 | 0.870 | 0.279 | 0.663 | 0.873 | 0.523 | 1.009 | 0.749 | 0.630 |
| MRIV (Frauen & Feuerriegel, 2022) | 0.579 | 0.631 | 0.760 | 0.189 | 0.586 | 1.091 | 0.471 | 0.880 | 0.669 | 0.556 |
| BASELINES (C): FOUNDATION MODEL-BASED | | | | | | | | | | |
| DoPFN (Robertson et al., 2025) | 0.471 | 0.528 | 0.787 | 0.322 | 0.649 | 1.723 | 0.416 | 0.588 | 0.722 | 0.545 |
| CausalFM (ours) | 0.515 | 0.600 | 0.704 | 0.152 | 0.538 | 0.934 | 0.414 | 0.826 | 0.600 | 0.509 |

Reported: PEHE (mean ± standard deviation.) Lower = better (best in bold).

## H.5 ADDITIONAL RESULTS FOR MISSPECIFICATION AND PRIOR DESIGN CHOICES

**Misspecification of causal settings.** We conduct experiments to study the sensitivity to using an incorrect identifiability strategy. Specifically, we generate data from (i) an IV SCM and (ii) a front-door SCM (as in our main experiments), and compare a model trained under the correct identifiability design (IV or front-door, respectively) with a model trained under an incorrect back-door design. The results are in Table 9. As expected, using a misspecified identifiability strategy consistently worsens the PEHE. This highlights the importance of including the correct identifiability assumption into the prior specification, which we propose for CausalFM.

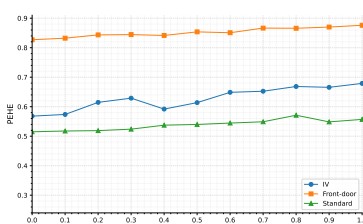

Figure 2: Robustness of our model to difference choices of the prior.

**Prior design choices.** We also analyze the robustness of our model to different choices of the prior. In particular, we vary the strength of unobserved confounding in the data-generating process, controlled by a parameter $\alpha \in [0, 1]$. The results in Fig. 2 show that our model remains robust as $\alpha$ increases. This again confirms the strong performance of CausalFM.

## H.6 COMPUTATIONAL TIME

We report computation time in this section. For our method and other foundation-model-based approaches, we show inference time since these models do not need fine-tuning after pretraining. For standard baselines, which must be trained for each dataset separately, we report the average total time per dataset, including both training and inference. As shown in Tables 10 and 11, our model is highly efficient.

Table 9: Analysis of the effect of misspecified identifiability strategies.

| Data Generating SCM | Strategy | Identifiability Used | PEHE |
|---|---|---|---|
| IV SCM | Correct | IV | 0.422 |
| | Incorrect | Back-door | 0.489 |
| Front-door SCM | Correct | Front-door | 0.847 |
| | Incorrect | Back-door | 0.876 |

Table 10: Overall time comparison for standard CATE setting.

| Method | Time (s) |
|---|---|
| BASELINES (A): STANDARD CATE ESTIMATORS | |
| S-learner (Künzel et al., 2019) | $2.76 \times 10^0$ |
| T-learner (Künzel et al., 2019) | $3.21 \times 10^0$ |
| TARNet (Shalit et al., 2017b) | $3.98 \times 10^0$ |
| DR-learner (Kennedy, 2023b) | $1.78 \times 10^1$ |
| RA-learner (Curth & van der Schaar, 2021) | $1.24 \times 10^1$ |
| X-learner (Künzel et al., 2019) | $1.93 \times 10^1$ |
| BASELINES (B): FOUNDATION MODELS-BASED METHODS | |
| CausalPFN (Balazadeh et al., 2025) | $1.27 \times 10^0$ |
| DoPFN (Robertson et al., 2025) | $2.31 \times 10^0$ |
| **CausalFM (ours)** | $4.90 \times 10^{-1}$ |

Lower = better. Reported: Time in seconds.

Table 11: Overall time comparison for IV setting.

| Method | Time (s) |
|---|---|
| BASELINES (A): STANDARD IV ESTIMATORS | |
| KIV (Singh et al., 2019) | $3.24 \times 10^{-1}$ |
| DRIV (Syrgkanis et al., 2019) | $3.87 \times 10^1$ |
| DeepIV (Hartford et al., 2017) | $1.27 \times 10^1$ |
| DeepGMM (Bennett et al., 2019) | $1.85 \times 10^1$ |
| DMLIV (Syrgkanis et al., 2019) | $1.85 \times 10^1$ |
| DFIV (Xu et al., 2021) | $1.74 \times 10^1$ |
| MRIV (Frauen & Feuerriegel, 2022) | $1.56 \times 10^1$ |
| BASELINES (B): FOUNDATION MODELS-BASED METHODS | |
| DoPFN (Robertson et al., 2025) | $6.53 \times 10^0$ |
| **CausalFM (ours)** | $4.72 \times 10^{-1}$ |

Lower = better. Reported: Time in seconds.

