# OpenReview forum: "Foundation Models for Causal Inference via Prior-Data Fitted Networks"
_ICLR.cc/2026/Conference — ICLR 2026 Poster_

### Official Review · Reviewer_V8FG · 2025-10-25

**Soundness:** 3
**Presentation:** 3
**Contribution:** 3
**Rating:** 8
**Confidence:** 4

**Summary:**

This paper introduces CausalFM, a framework for training transformer-based Prior-data Fitted Networks (PFNs) as foundation models for causal inference. The approach centers on "C-SCM-Priors," novel prior distributions over Structural Causal Models (SCMs) constructed to explicitly satisfy identifiability assumptions for specific causal settings (e.g., back-door, front-door, IV). The paper presents a theoretical argument that restricting the prior to identifiable SCMs is necessary for the resulting PFN to be "well-specified" and yield consistent causal effect estimates. CausalFM is pre-trained on synthetic data from these priors, enabling it to perform Bayesian causal inference (specifically, CATE estimation) via in-context learning on new observational datasets without retraining. Experiments demonstrate competitive performance against specialized baselines in the targeted settings.
[Note: I have used LLMs to improve my writing and help me answer paper questions]

**Strengths:**

- The paper provides a clear theoretical argument for why PFN priors in this Bayesian approximation context should enforce identifiability for consistent estimation (I am a little skeptical if we should make this assumption, see below - but realizing and demonstrating it is very valuable).
- Introduces a structured way to build priors over SCMs using BNNs that respect the assumed causal graph structure and identifiability conditions.
- CausalFM is designed to handle back-door, front-door, and IV settings, offering broader applicability than methods restricted to unconfoundedness, like CausalPFN.
- Achieves strong average empirical results across multiple benchmarks and settings compared to specialized estimators that require per-dataset training and tuning.

**Weaknesses:**

- Reliance on Correct Identifiability Assumptions: The framework's core premise requires the user to correctly identify the true causal structure and select the appropriate identifiability strategy (back-door, front-door, IV) before applying the model. This is a strong assumption, as determining the correct causal graph and valid adjustment strategy from domain knowledge alone is often a major challenge in real-world applications. CausalFM automates estimation given these assumptions but offers no mechanism to validate them or handle uncertainty about the true causal structure. Potential Mismatch with Reality vs. Uncertainty Modeling: By strictly requiring identifiable priors for consistency (Thm 4.3), CausalFM might struggle or produce overly confident (but potentially wrong) estimates if the real-world DGP is non-identifiable or if the chosen identifiable structure is incorrect. CausalFM prioritizes potential consistency under strong assumptions over explicitly modeling uncertainty arising from potential non-identifiability.

- The effectiveness depends on the specific design of the BNN-based C-SCM-Prior. The paper lacks sensitivity analyses regarding the prior's construction (e.g., BNN architecture, diversity of sampled SCMs) and how well this synthetic prior generalizes to the complexities of real-world data generation processes.

**Questions:**

- How sensitive is CausalFM's performance if the user specifies a causal setting (e.g., assumes back-door identifiability) but the underlying data violates this assumption mildly (e.g., weak unobserved confounding)? Does the model fail catastrophically, or does it offer graceful degradation?
- How critical are the specific choices made in the BNN architecture and sampling process for the C-SCM-Prior (Sec 4.3)? Were alternatives explored, and how much does performance vary with changes to the prior generation mechanism?

---

> ### Author Response · Authors · 2025-11-19
> **Response to Reviewer V8FG (1)**
>
> Thank you very much for your thoughtful review\! Below, we address all your comments and suggestions. We incorporated all points marked with **Action** into the revised version of our paper (marked in $\\color{blue}{\\text{blue}}$).
>
> ## Response to Weaknesses
>
> * **Reliance on correct identifiability assumptions:** Thank you for bringing up this important point. You are correct that one of our main motivations for CausalFM (as compared to e.g., Do-PFN \[1\]) is *to separate the identifiability and estimation process*. Our reasoning is as follows:
>   * **Asymptotically unbiased causal inference.** As we show in Theorem 4.3, incorporating identifiability assumptions into the prior is **necessary** for asymptotically unbiased causal inference. As a consequence, methods that ignore identifiability assumptions yield biased causal effect estimates, even if we collect large amounts of data. This is highly undesirable in practice.
>   * **Informative predictive-posterior distributions.** If we do not impose any assumption on the DGP, it is well known that causal effect estimation is not just fundamentally biased, **but also that this bias can be of arbitrary size.** For example, the backdoor-adjustment bias due to omitted unobserved confounding can be written in closed form depending on confounding strength (see, e.g., \[2, 3\]). Thus, if the PFN-prior assigns positive probability mass for DGPs with arbitrary confounding strength, the predictive-posterior must respect the possibility of arbitrarily biased treatment effects, thus rendering PFN-based inference completely noninformative.
>   * **Clear separation between domain knowledge and statistical inference.** One might argue that a possible remedy would be to restrict the PFN prior only to DGPs with somewhat “weak” identifiability violations (e.g., weak unobserved confounding). However, we argue that this would correspond to assumptions/ domain knowledge on the DGP, similar to those in our paper, that must be made transparent for practitioners and could also possibly be violated.
>
>     In short, **our paper follows established causal inference philosophy and separates identifiability from estimation**: the identifiability step (choosing the causal setting) requires careful modeling and usage of domain knowledge, while the estimation step can be handed over to our CausalFM. If practitioners suspect identifiability assumptions may be violated, we recommend performing causal sensitivity analysis \[2, 3\] to assess the extent of potential violations.
>
> * **Sensitivity analysis for Prior construction.** Thank you for your input. To avoid redundancies, we kindly refer you to our second response to “Questions” below.

---

> ### Author Response · Authors · 2025-11-19
> **Response to Reviewer V8FG (2)**
>
> ## Response to Questions
>
> * **Misspecification of causal settings.** Misspecification of causal inference settings has been extensively studied in the literature. Such misspecification generally introduces bias, **which is independent of the estimation method used for statistical inference**. As an example, consider a ground-truth DGP with unobserved confounding (e.g., IV setting), and we mistakenly specify an unconfounded backdoor adjustment setting. Then, the confounding bias can be quantified in closed form depending on the strength of unobserved confounding (see e.g.,  \[2, 3\]). Moreover, any estimation method (e.g., our CausalFM, CausalPFN, or standard causal inference estimators) would suffer from this same bias, no matter the size of the available training data. As a consequence, **the question regarding sensitivity to misspecification is not specific to our method, but to any causal inference estimator employed in this misspecified setting.**
>
>   Nevertheless, we acknowledge the importance of empirically reporting the sensitivity to causal misspecification. **Action:**  We conduct **new experiments** to study the robustness to using an incorrect identifiability strategy (see **new Table 7** and **new Sec.6.5**). Specifically, we generate data from (i) an IV SCM and (ii) a front-door SCM (as in our main experiments), and compare a model trained under the correct identifiability design (IV or front-door, respectively) with a model trained under an incorrect back-door design. As expected, using a misspecified identifiability strategy consistently worsens PEHE, confirming that misspecification of the causal setting leads to biased estimates. This highlights the importance of including the correct identifiability assumption into the prior specification, which we propose for CausalFM.
>
>
>
>
> * **Prior design choices.** Thank you for this important question. We agree that additional analysis regarding prior design choices will help to strengthen our contributions. **Action:**  We thus performed **new experiments** to analyze the robustness of our model to changes in the prior (see **new Fig. 2 in Appendix H.5**). In particular, we vary the strength of unobserved confounding in the data-generating process, controlled by a parameter $\\alpha \\in \[0, 1\]$. **The experiments show that our model remains robust in correctly identified settings** as $\\alpha$ increases.
>
> References
> \[1\] Robertson et al. (2025). Do-PFN: In-Context Learning for Causal Effect Estimation. NeurIPS.
> \[2\] Dorn et al. (2024). Doubly-Valid/Doubly-Sharp Sensitivity Analysis for Causal Inference with Unmeasured Confounding. JASA.
> \[3\] Frauen et al. (2023). Sharp Bounds for Generalized Causal Sensitivity Analysis. NeurIPS.

---

> > ### Comment · Reviewer_V8FG · 2025-11-26
> > **All issues have been answered**
> >
> > Thank you for the detailed and clear review, which has answered my open points.

---

### Official Review · Reviewer_Zj58 · 2025-11-01

**Soundness:** 3
**Presentation:** 4
**Contribution:** 3
**Rating:** 6
**Confidence:** 4

**Summary:**

In this paper, the authors propose CausalFM for training foundation models for causal inference based on existing Prior-data Fitted Networks (PFNs). The core idea is to pre-train a transformer model on synthetic data generated from a family of Structural Causal Model (SCM), enabling zero-shot causal estimation in various settings, including back-door, front-door, IV, via in-context learning without retraining. The approach addresses a significant limitation of traditionally causal inference methods. The empirical results show competitive performance.

**Strengths:**

Beyond dataset-specific causal inference to in-context causal inference is a promising and important research direction.  In the paper, the authors propose a framework which is not restricted to a single causal setting. It supports back-door, front-door, and instrumental variable scenarios within a single model.

The introduction of the "well-specified prior" concept and the argument for incorporating identifiability assumptions into the prior are valuable theoretical insights.

The results on synthetic, semi-synthetic (Jobs), and other datasets demonstrate the model's versatility and strong performance.

**Weaknesses:**

The pre-training methodology in this paper appears largely similar to exsiting work in this area

The proof sketches are quite dense and lacks intuition, then are hard to follow.

The 24-hour training time on an A100 GPU is mentioned, but there is no discussion of inference speed or comparative training costs of the baselines.

**Questions:**

In fact, some researchers have begun leveraging PFNs to pre-train causal foundation models, such as the following work. The pretraining method used in this paper is very simimlar to the existing work, although the authors have discussed these methods in the paper. In addition, the authors critique the following work for being restricted to back-door adjustment. However, this work can actually handle back-door, front-door, and IV settings via in-context learning without retraining. Since the pre-training methodology in this paper appears largely similar, its claimed novelty over these prior works remains unclear.

Jake Robertson, Arik Reuter, Siyuan Guo, Noah Hollmann, Frank Hutter, and Bernhard Scholkopf. Do-pfn: In-context learning for causal effect estimation, arXiv preprint, arXiv:2506.06039, 2025.

In the back-door training loss (Eq. 9), the model is trained to predict the individual treatment effect directly. What is the rationale behind this and whether you experimented with predicting potential outcomes Y(a) separately and then differencing?

The prior in Sec. 4.3 is based on a well-specified C-DAG provided by the practitioner. How sensitive is the method to misspecification of this C-DAG?

---

> ### Author Response · Authors · 2025-11-19
> **Response to reviewer Zj58 (1)**
>
> Thank you very much for your positive review\! Below, we address your comments and suggestions in detail. We incorporated all points marked with **Action** into the revised version of our paper (marked in $\\color{blue}{\\text{blue}}$).
>
> ## Response to Weaknesses
>
> * **Methodological novelty:** Thank you for allowing us to elaborate on the novelty of our pre-training methodology as compared to existing works. **The only two existing PFNs for causal inference that we are aware of are (i) CausalPFN \[1\] and (ii) Do-PFN \[2\].**
>   (i) At a high level, **CausalPFN can be viewed as a special case of our CausalFM framework** for the (unconfounded) backdoor adjustment setting (ignoring differences in prior specification and model architecture). As such, CausalPFN does not support causal inference settings beyond backdoor adjustment and will provide biased estimates, e.g., in settings with unobserved confounding.
>   (ii) The paper you mentioned, Do-PFN, takes a different approach: As you mentioned correctly, it pre-trains the PFN on all kinds of causal inference settings beyond backdoor adjustment. The main difference to our CausalFM is that **Do-PFN does not restrict its prior to identifiable causal inference settings.** As we show in Theorem 4.3, **Do-PFN can yield asymptotically biased causal effect estimates, which is highly undesirable**. In practice, this can mean that, even for large datasets, Do-PFN provides non-informative predictive-posterior distributions. An example: If the Do-PFN prior puts probability mass on DGPs with strong unobserved confounding (without IVs, etc.), the posterior distribution must respect this, and inconclusive results on *any* dataset (which is also reflected in our experiments). Of course, this could be softened by restricting the prior to allow for only limited confounding strength (similar to causal sensitivity analysis \[3, 4\]), but this would be somewhat arbitrary and would need to be justified by domain knowledge. In short, **our paper follows established causal inference philosophy and separates identifiability from estimation**: the identifiability step (choosing the causal setting) requires careful modeling and usage of domain knowledge, while the estimation step can be handed over to our CausalFM.
> * **Intuition for proofs:** Thank you for pointing this out. **Action:** We made the proof of Theorem 4.3 more elaborate and added clarifications and explanations to improve readability.
> * **Reporting of inference times:** Thank you for your suggestion. **Action:** We appreciate this point and have **added a comparison of the** **time cost** to our pape**r**   (new **Table 5 & 6** and new **Sec. 6.4**). Therein, we also report the time cost of the baselines for comparison.

---

> ### Author Response · Authors · 2025-11-19
> **Response to reviewer Zj58 (2)**
>
> ## Response to Questions
>
> * **Comparison with Do-PFN:** To avoid repetition, we kindly refer to our first “Response to Weaknesses”. In particular, the **key difference between CausalFM and Do-PFN is the separation of causal identification and estimation**, and thus ingraining identifiability assumptions into our CausalFM prior.
> * **Treatment effect vs potential outcomes:** Thank you for bringing up this important point\! In principle, nothing stops our CausalFM from modeling potential outcomes $Y(a)$ instead of treatment effects $Y(1) \-  Y(0)$ (our methodology is compatible with both). In practice, however, practitioners are arguably often more interested in treatment effects to inform treatment decisions. While we could use potential outcome predictions $\\hat{Y}(a)$ for treatment effect predictions via the plug-in estimator $\\hat{Y}(1) \- \\hat{Y}(0)$, this approach is known to suffer from plug-in bias and thus is suboptimal \[5\]. As a consequence, direct modeling of treatment effects is always preferred if the ultimate goal is to obtain treatment effects.
> * **Misspecification of C-DAG:** Misspecification of causal inference settings (or C-DAGs) has been extensively studied in the literature. Such misspecification generally introduces bias, **which is independent of the estimation method used for statistical inference**. As an example, consider a ground-truth DGP with unobserved confounding (e.g., IV setting), and we mistakenly specify an unconfounded backdoor adjustment setting. Then, the confounding bias can be quantified in closed form depending on the strength of unobserved confounding (see, e.g.,  \[3, 4\]). Moreover, any estimation method (e.g., our CausalFM, CausalPFN, or standard causal inference estimators) would suffer from this same bias, no matter the size of the available training data. As a consequence, **the question regarding sensitivity to misspecification is not specific to our method, but to any causal inference estimator employed in this misspecified setting.**
>
>   Nevertheless, we acknowledge the importance of empirically reporting the sensitivity to causal misspecification. **Action:** We conduct **new experiments** to study the sensitivity to using an incorrect identifiability strategy (see  **new Table 7** and new **Sec.6.5**). Specifically, we generate data from (i) an IV SCM and (ii) a front-door SCM (as in our main experiments), and compare a model trained under the correct identifiability design (IV or front-door, respectively) with a model trained under an incorrect back-door design. As expected, using a misspecified identifiability strategy consistently worsens PEHE, confirming that misspecification of the causal setting leads to biased estimates. This highlights the importance of including the correct identifiability assumption into the prior specification, which we propose for CausalFM.
>
>
> ## References
>
> \[1\] Balazadeh et al. (2025). CausalPFN: Amortized Causal Effect Estimation via In-Context Learning. NeurIPS.
> \[2\] Robertson et al. (2025). Do-PFN: In-Context Learning for Causal Effect Estimation. NeurIPS.
> \[3\] Dorn et al. (2024). Doubly-Valid/Doubly-Sharp Sensitivity Analysis for Causal Inference with Unmeasured Confounding. JASA.
> \[4\] Frauen et al. (2023). Sharp Bounds for Generalized Causal Sensitivity Analysis. NeurIPS.
> \[5\] Kennedy (2022). Semiparametric doubly robust targeted double machine learning: a review. Handbook of Statistical Methods for Precision Medicine.

---

> > ### Comment · Reviewer_Zj58 · 2025-11-28
> >
> > This is a new area for causal inference and thanks for your clarification. The authors should clearly clarify the differences between CausalFM and the existing work of CausalPFN, DO-PFN, expecially with the recent DO-PFN to enhance their contributions, since the pre-training methodology of CausalFM, CausalPFN, DO-PFN seems very similar.

---

> > > ### Author Response · Authors · 2025-11-29
> > >
> > > Thank you for your response. Below, we address all your comments.
> > >
> > > **1\. Clarification: Differences between CausalFM (ours), CausalPFN[1], and Do-PFN[2]**
> > >
> > > While all three methods leverage PFN-style pre-training on synthetic data, **CausalFM is fundamentally different in purpose, prior construction, and theoretical guarantees**.
> > >
> > > **(1) CausalFM separates *identifiability* from *estimation*.**
> > >
> > >  The central motivation of CausalFM is that causal identification (choosing an identifiable setting such as back-door, IV, or front-door) must be handled *before* estimation. This mirrors classical causal inference practice and ensures that the PFN only learns within an **identifiable** causal setting. Our reasoning for identifiability is as follows:
> > >
> > >   * **Asymptotically unbiased causal inference.** As we show in Theorem 4.3, incorporating identifiability assumptions into the prior is **necessary** for asymptotically unbiased causal inference. As a consequence, methods that ignore identifiability assumptions yield biased causal effect estimates, even if we collect large amounts of data. This is highly undesirable in practice.
> > >   * **Informative predictive-posterior distributions.** If we do not impose any assumption on the DGP, it is well known that causal effect estimation is not just fundamentally biased, **but also that this bias can be of arbitrary size.** For example, the backdoor-adjustment bias due to omitted unobserved confounding can be written in closed form depending on confounding strength (see, e.g., \[3, 4\]). Thus, if the PFN-prior assigns positive probability mass for DGPs with arbitrary confounding strength, the predictive-posterior must respect the possibility of arbitrarily biased treatment effects, thus rendering PFN-based inference completely noninformative.
> > >   * **Clear separation between domain knowledge and statistical inference.** One might argue that a possible remedy would be to restrict the PFN prior only to DGPs with somewhat “weak” identifiability violations (e.g., weak unobserved confounding). However, we argue that this would correspond to assumptions/ domain knowledge on the DGP, similar to those in our paper, that must be made transparent for practitioners and could also possibly be violated.
> > >
> > > In short, **our paper follows established causal inference philosophy and separates identifiability from estimation**: the identifiability step (choosing the causal setting) requires careful modeling and usage of domain knowledge, while the estimation step can be handed over to our CausalFM. If practitioners suspect identifiability assumptions may be violated, we recommend performing causal sensitivity analysis \[3, 4\] to assess the extent of potential violations.
> > >
> > > In contrast:
> > >
> > > * **CausalPFN** \[1\] assumes *only* back-door adjustment and therefore cannot handle IV or front-door, resulting in bias under unobserved confounding.
> > >
> > > * **Do-PFN** \[2\] mixes causal graphs in a single prior without conditioning on which setting is identifiable, which, as our Theorem 4.3 shows, can lead to asymptotically biased estimates and non-informative posteriors.
> > >
> > > **(2) This philosophy requires a fundamentally different *prior construction*.**
> > >  Because identifiability is encoded at the level of causal structure, CausalFM introduces **C-SCM priors** and **C-DAGs** that enforce the assumptions required by each identifiability strategy.
> > >
> > > Do-PFN [2] does **not** encode identifiability constraints into the prior family for their *prior construction*. CausalPFN [1] is **restricted** solely to back-door adjustment.
> > >
> > > **(3) Beyond this framework, we contribute a new theory showing that identifiability must be incorporated into PFN priors.**
> > >  Theorem 4.3 proves that if a PFN prior places nonzero mass on SCMs that violate the identifiability conditions of the chosen setting, then the resulting posterior predictive interventional distribution is necessarily misspecified and cannot yield consistent causal effect estimates, even with infinite data.
> > >
> > > This explains the empirical behavior of Do-PFN, which may return non-informative posteriors when its prior includes SCMs with strong unobserved confounding and no valid instruments. Our theoretical results show that this issue is structural, *not merely* an implementation detail.
> > >
> > > **(4) Empirically,** our model outperforms others in different settings. Besides, we also have experiments showing the necessity to have the correct identifiability assumption in the prior specification.
> > >
> > > **Action:  We added this clarification to our paper (see Appendix A.1).**

---

> > ### Comment · Reviewer_Zj58 · 2025-11-28
> >
> > Additionally, please clarify the input format for CausalFM. Specifically, does it require an entire dataset with n samples and m variables, or can it process a single data sample? A clarification on this point would be greatly appreciated.

---

> ### Author Response · Authors · 2025-11-29
>
> **2\. Clarification on input format**
>
> CausalFM operates as an in-context learner like other foundation models, meaning it approximates Bayesian inference by conditioning on a dataset provided in its context window. Therefore, it requires a dataset to make predictions for specific query samples.
>
> \- **Input Structure**: The model accepts a dataset $\\mathcal{D}\_n=\\left\\{\\left(x\_i, a\_i, y\_i\\right)\\right\\}\_{i=1}^n$ acting as the context (or support set) and a query point $x\_{\\text {query }}$ (or a batch of query points).
>
> \- **Mechanism**: The transformer processes the entire sequence of observed data $\\mathcal{D}\_n$ using self-attention to extract context-dependent representations. It then outputs the posterior predictive distribution for the causal quantity (e.g., CATE given the context $\\mathcal{D}\_n$ and the specific query $x\_{\\text {query }}$.
>
> \- **Comparison to fine-tuning baselines:** In practice, standard baselines require an explicit training phase on a training set before evaluation. In contrast, CausalFM takes the "training" data as the input context (support set) and directly generates predictions for the test data (query set) in a single forward pass.
>
>
>
>
> **Action:  We added this clarification to our paper (see Appendix D.1).**
>
> ## References
>
> \[1\] Balazadeh et al. (2025). CausalPFN: Amortized Causal Effect Estimation via In-Context Learning. NeurIPS.
> \[2\] Robertson et al. (2025). Do-PFN: In-Context Learning for Causal Effect Estimation. NeurIPS.
> \[3\] Dorn et al. (2024). Doubly-Valid/Doubly-Sharp Sensitivity Analysis for Causal Inference with Unmeasured Confounding. JASA.
> \[4\] Frauen et al. (2023). Sharp Bounds for Generalized Causal Sensitivity Analysis. NeurIPS.

---

### Official Review · Reviewer_r9r3 · 2025-11-02

**Soundness:** 3
**Presentation:** 3
**Contribution:** 3
**Rating:** 6
**Confidence:** 5

**Summary:**

This paper proposes CausalFM, a new framework for training foundation models (transformers) to perform causal inference via prior-data fitting. The authors formalize how to construct Bayesian prior distributions based on structural causal models (SCMs) for various causal inference tasks, and derive necessary criteria to ensure these priors lead to valid, identifiable causal inferences. Using this theory, they propose a novel family of SCM-based priors implemented through Bayesian neural networks, which enables the PFN (Prior-Data Fitted Network) to carry out Bayesian causal effect estimation in-context for settings including back-door (unconfounded observational studies), front-door (mediator adjustment), and instrumental variable (IV) scenarios. The paper then instantiates CausalFM by training transformer models on synthetic data generated from those SCM-based priors, effectively teaching the model to infer causal effects (e.g. Conditional Average Treatment Effect, CATE) without retraining on new datasets. Empirically, CausalFM achieves competitive or superior performance compared to both standard specialized estimators and prior foundation-model baselines, across diverse benchmarks for CATE (unconfounded), IV, and front-door settings. In many cases, the CausalFM model’s in-context predictions match or outperform state-of-the-art alternatives even though those baselines are trained specifically for each task. Overall, the paper’s main result is a general recipe to train a single tabular foundation model for causal inference tasks, offering a new paradigm that could improve flexibility (test-time inference without retraining) in fields like medicine and economics.

**Strengths:**

- The paper introduces a novel approach by leveraging prior-data fitted networks for causal inference, which might be the first to provide a comprehensive foundation model (CausalFM) covering multiple settings in one framework. Unlike past works that focus on a single identification strategy (e.g. only back-door criteria) or require task-specific models, CausalFM uses SCM-based priors to train a transformer that can flexibly handle back-door, front-door, and IV adjustments within one model. This is one step closer to a real causal FM: the model effectively learns to “select” the appropriate causal estimation formula from data context, enabling zero-shot inference on new datasets without retraining – a clear paradigm shift from traditional retrain-per-dataset approaches. The introduction of SCM-guided synthetic training data (via a Bayesian neural network prior) is especially novel, as it injects domain causal knowledge into the foundation model’s pretraining process. This combination of foundation models + causal SCM priors is an original contribution that extends the foundation model concept into the causal inference domain in a principled way.

- The paper is grounded in strong theory. The authors formally define conditions under which a causal effect is identifiable and incorporate those into the prior design. In particular, they derive necessary criteria for a valid causal prior, proving that any well-specified prior must assign zero probability to SCMs that violate the identifiability of the target causal query. This result (Theorem 4.3) ensures that the model’s Bayesian posterior cannot mislead us with non-identifiable alternatives, thereby guaranteeing asymptotic consistency of the causal estimates under the given assumptions. Furthermore, by building on a Bayesian PFN framework, CausalFM naturally provides uncertainty quantification and connects to Bayesian consistency results (e.g. invoking a Bernstein–von Mises argument for the posterior on the observational distribution). The paper’s separation of identifiability (ensured via SCM prior design) and statistical estimation (learned by the PFN) follows established causal inference principles (Pearl’s identification-before-estimation philosophy), lending the approach a sound theoretical foundation. Overall, the inclusion of formal definitions, theorems, and proofs about prior construction, consistency, and identifiability demonstrates a high level of theoretical rigor that bolsters confidence in the method’s validity.

- Strong Empirical Performance: The experimental evaluation is thorough and shows competitive or superior performance of CausalFM on a wide range of benchmarks. The authors test their model in standard back-door (observational CATE estimation) scenarios (including synthetic datasets and a semi-synthetic Jobs dataset), in instrumental variable settings (with both binary and continuous instruments), and in front-door mediation scenarios. Across these experiments, CausalFM’s one-model approach achieves error rates (PEHE) that are on par with or better than specialized models. For example, in CATE estimation, CausalFM slightly outperforms several state-of-the-art estimators (e.g. S-learner, T-learner, TARNet) and also improves over prior foundation model baselines: CausalFM attained a lower PEHE (≈0.51) than CausalPFN (≈0.56) or DoPFN (≈0.59) on synthetic CATE benchmarks. In the IV setting, CausalFM similarly achieved the best PEHE among foundation models (0.42 vs. 0.52 for DoPFN) and was comparable to the top specialized IV methods. Notably, in a challenging front-door adjustment task with confounded treatment and mediator, the CausalFM model obtained a PEHE around 0.90, outperforming a baseline foundation model (DoPFN at 1.27) and all but the very best plug-in front-door estimator. These results indicate that CausalFM not only generalizes across different causal inference problems but often matches or exceeds the accuracy of dedicated models, all while requiring no retraining per new dataset. The consistency of CausalFM’s performance across diverse settings and its edge over previous PFN approaches (which were limited to back-door or had no identifiability guarantees) stand out as a major strength. This empirical evidence convinces the reader that the proposed foundation model approach is viable and competitive in practice.

**Weaknesses:**

- While CausalFM is conceptually flexible, there may be practical scalability challenges. Training a PFN of this sort involves generating and learning from a very large number of synthetic datasets drawn from complex SCM priors, which is computationally intensive. The transformer model itself has a fixed context length and model size – this could limit the scale of datasets it can handle at test time (e.g. number of samples or covariates) unless the architecture is scaled up. The paper does not extensively discuss memory or runtime implications; however, foundation models for tabular data (like TabPFN) are known to handle only modest-sized datasets due to the need to input the dataset as context. If one were to apply CausalFM to truly large real-world datasets (say, thousands of units or very high-dimensional covariates), it is unclear if the current model would remain efficient or accurate. In short, questions of scalability (both in training cost and inference on large data) remain a concern. The method’s impressive performance is demonstrated on reasonably sized benchmarks, but its feasibility on substantially bigger or more complex tasks has not been shown, potentially limiting its immediate practical adoption for “big data” causal inference.

- The work would benefit from deeper analysis of why CausalFM works so well. Currently, there is little in the way of ablation studies or interpretability insights. For example, the framework involves several design choices – using a Bayesian neural network prior, simulating counterfactuals during training, particular choices in the SCM parameterization – yet the paper does not present ablation experiments to quantify the contribution of each component (e.g., what if the prior did not enforce certain assumptions, or if the counterfactual simulation was removed?). Without such ablations, it is hard to assess which aspects of the CausalFM design are most critical. Additionally, the interpretability of the PFN’s in-context inference mechanism is not explored. CausalFM essentially acts as a black-box that implicitly decides which causal adjustment to apply (back-door vs front-door vs IV) based on the input data. However, the readers are not shown how the model makes this decision or whether it aligns with known causal formulas in each case. There is no analysis, for instance, of attention weights or internal representations to reveal if the model focuses on certain variables (e.g. an instrument or mediator) to choose the appropriate formula. This lack of interpretability means practitioners might find it hard to trust or validate the model’s reasoning on a given dataset. In summary, the paper’s experiments focus on accuracy, but omit diagnostic analyses: one misses ablation studies to justify design choices and interpretability checks to illuminate the model’s decision-making. This is a limitation because understanding the model’s behavior would greatly enhance confidence and provide insights for future improvements.

- The success of CausalFM hinges on the assumption that the chosen prior distribution (the family of SCMs used in training) accurately captures the relevant causal structure of the target domain. This could make the approach sensitive to mis-specification. If a real-world scenario violates the assumptions baked into the prior (for example, there is an unobserved confounder not accounted for, or the functional relationships differ substantially), the model’s inferences may become unreliable. The authors explicitly restrict the priors to settings where identifiability holds, which is sensible, but it also means the model is not trained to handle scenarios outside those assumptions (e.g. it wouldn’t know how to behave if confronted with a non-identifiable case, except to potentially output high uncertainty). It would strengthen the work to understand how robust the model is to slight violations of its assumptions – this is not addressed in the current evaluation. Moreover, real-world validation is limited in the experiments. While the paper includes a semi-synthetic example (Jobs dataset with simulated outcomes) and numerous fully synthetic benchmarks, it does not demonstrate CausalFM on purely real observational data where ground-truth causal effects are unknown. Consequently, it remains to be seen how the model performs in practice on real datasets, and how one would verify its estimates when the true effect is not available for comparison. The lack of real-world case studies means the approach’s practical utility is still somewhat speculative. In essence, the framework’s generalizability to real, messy data and its robustness to deviations from assumed priors are open questions not fully answered by the paper. These limitations suggest that further work is needed to test CausalFM under less ideal conditions and to guide users in setting an appropriate prior for their specific application.

**Questions:**

- Robustness to Prior Mis-specification: How sensitive is CausalFM to the correctness of its prior assumptions in practice? In a scenario where the true data-generating process deviates from the assumed SCM prior (for instance, an unobserved confounder is present when the model assumed none, or the functional form of relationships is different), what would happen to the model’s performance? Would the in-context learner recognize the model mismatch (perhaps via larger predictive uncertainty), or could it yield biased estimates? It would be helpful if the authors could clarify whether any experiments were done to assess robustness when the test data falls slightly outside the support of the training prior, and if not, how might one ensure reliability of CausalFM in such cases (e.g., by broadening the prior or diagnosing posterior outputs).

- Interpretability and Practical Use: While the model automatically selects an identification strategy based on the data, it operates as a black box from the user’s perspective. Do the authors have insights or plans for making CausalFM’s reasoning more interpretable? For example, is it possible to extract which causal adjustment the model is implicitly using for a given dataset (such as detecting that it’s performing an IV-like computation versus a back-door adjustment)? Understanding this could be important for practitioners to trust the results. Moreover, how would the authors recommend validating CausalFM’s outputs on a real-world problem where the true causal effect is unknown? Are there diagnostic checks or visualizations one can use (perhaps analyzing the learned attention weights or the model’s posterior over causal effects) to ensure the model’s inference aligns with domain knowledge? Clarifying these points would help users apply CausalFM more confidently and transparently in practice.

---

> ### Author Response · Authors · 2025-11-19
> **Response to Reviewer r9r3 (1)**
>
> Thank you for your positive review and your helpful comments. Below we have drafted careful responses to your suggestions. We incorporated all points marked with **Action** into the revised version of our paper (marked in $\\color{blue}{\\text{blue}}$).
>
> ## Response to Weaknesses
>
> * **Scalability to large datasets:** Thank you for bringing up this important point. You are right, this is a well-known restriction of PFNs. Our current implementation aims to showcase the feasibility of causal foundation models for modest sample sizes. For our experiments, we used 10k datasets. However, we are confident that this can be scaled up in future work. PriorLabs \[1\] is currently working on promising approaches to scale up PFNs (up to 100k sample size), and we are confident that similar approaches could be used to scale up CausalFM. We still expect our CausalFM to be relevant in many real-world applications, where datasets are often of similar or even smaller size.
>
>   **Runtime**: We appreciate this point and have **added a comparison of the time cost report** to our paper (see **new Tables 5 & 6** and **new Sec. 6.4** ). Therein, we also report the time cost for the baselines.
>
>
> * **Ablation studies and model interpretability:** Thank you for pointing this out. Adding interpretability to our models (e.g., using specialized architectures or post-hoc explainability methods) is definitely an interesting path for future improvements, though we believe that this goes beyond the scope of our work. Nevertheless, we agree that it is important to add additional insights into the drivers of CausalFM’s performance. **Action:** We discuss potential directions for future work such as improving interpretability (see our **revised Section 6**).
>
> * **Prior misspecification due to non-identifiability:** Misspecification of causal inference settings has been extensively studied in the literature \[2, 3\] and is not unique to our paper. Such misspecification generally introduces bias, **which is independent of the estimation method used for statistical inference**. As an example, consider a ground-truth DGP with unobserved confounding (e.g., IV setting), and we mistakenly specify an unconfounded backdoor adjustment setting. Then, the confounding bias can be quantified in closed form depending on the strength of unobserved confounding (see, e.g.,  \[2, 3\]). Moreover, ***any*** estimation method (e.g., our CausalFM, but also CausalPFN or any standard causal inference estimators) would suffer from this same bias, no matter the size of the available training data. As a consequence, **the question regarding sensitivity to misspecification is not specific to our method, but to any causal inference estimator employed in this misspecified setting.**
>
> * **Real-world experiments:** Thank you for the input. We would like to emphasize that proper causal evaluation is only possible on synthetic or semi-synthetic data, where counterfactual outcomes are accessible. Consequently, treatment effects on real-world data are unknown, and evaluating causal inference methods is challenging even if the treatment is randomized. Hence, our evaluation is consistent with large parts of the causal inference literature by using mostly synthetic and semi-synthetic data \[2, 3\].
>
>   That said, this challenge reflects a broader issue in causal inference \[4\]: translating methods validated on controlled or simulated data to real-world settings, where counterfactuals are unobservable and interventions are costly. A natural next step would be to rigorously evaluate our method in an applied A/B experimental setup to assess its empirical performance and robustness under real-world conditions. We expect such an evaluation to be particularly rewarding in domains like marketing or personalized recommendation, where treatments correspond to interventions such as targeted advertisements or incentives. We consider this a promising direction for future work. **Action:** We discuss potential directions for future work such as rigorously benchmarking CausalFM, CausalPFN, and standard causal inference estimators in an A/B experiments from the field such as in marketing (see our **revised Section 6**).

---

> ### Author Response · Authors · 2025-11-19
> **Response to Reviewer r9r3 (2)**
>
> ## Response to Questions
>
> * **Robustness to identifiability violations:** Thank you for bringing this up. As mentioned in our response to “Weaknesses”, robustness w.r.t. identifiability assumptions **is not specific to our method, but to any causal inference estimator employed in this misspecified setting.** Nevertheless, we acknowledge the importance of empirically reporting the sensitivity to causal misspecification.
>
>   We conduct **new experiments** to study the robustness of CausalFM to using an incorrect identifiability strategy (see **new Table 7** and **new Sec. 6.5**). Specifically, we generate data from (i) an IV SCM and (ii) a front-door SCM (as in our main experiments), and compare a model trained under the correct identifiability design (IV or front-door, respectively) with a model trained under an incorrect back-door design. As expected, using a misspecified identifiability strategy consistently worsens PEHE, confirming that misspecification of the causal setting leads to biased estimates. This highlights the importance of including the correct identifiability assumption into the prior specification, which we propose for CausalFM.
>
>
>
>   **Action:** We added the new experimental results under an incorrect identifiability strategy (see **new Table 7** and **new Sec. 6.5**).
>
>
>
>
> * **Model interpretability:** Thank you for the interesting question. As described in our response to “Weaknesses”, we believe that adding interpretability to our CausalFM is an important but non-trivial direction for future research. We thus discuss potential strategies in our revised **Sec. 6**.
>
>
> ## References
>
> \[1\] PriorLabs (2025). TabPFN-2.5: Advancing the State of the Art in Tabular Foundation Models. ArXiv.
> \[2\] Dorn et al. (2024). Doubly-Valid/Doubly-Sharp Sensitivity Analysis for Causal Inference with Unmeasured Confounding. JASA.
> \[3\] Frauen et al. (2023). Sharp Bounds for Generalized Causal Sensitivity Analysis. NeurIPS.
> \[4\] Poinsot et al (2025). Position: Causal Machine Learning Requires Rigorous Synthetic Experiments for Broader Adoption. ICML.

---

### Author Response · Authors · 2025-11-19
**Response to all reviewers**

Thank you very much for the positive evaluation of our paper and your helpful feedback\! We addressed all of them in the comments below. We also uploaded a revised version of our paper, highlighting key changes in $\\color{blue}{\\text{blue}}$.

Following the feedback of the reviewers, our **main improvements** are the following:

* **More discussion motivating the CausalFM design.** We clarified our motivation for separating identifiability and estimation in our CausalFM framework (see our **new discussion** in **Section 4.3**). In particular, we show that this separation follows established causal inference philosophy, allows for asymptotically unbiased causal inference, and a clear separation of assumptions/ domain knowledge and statistical inference.
*  **New experiments**. We conduct new experiments to analyze the robustness of CausalFM to an incorrect identifiability strategy (see **new Table 7** and **new Sec. 6.5**). We further analyze the robustness of CausalFM to different design choices of the prior (see **new Sec. 6.5**; and n**ew Fig. 2 in Appendix H.5**). These results highlight that (i) incorporating the correct identifiability assumptions into PFN priors is crucial for causal inference, and (ii) given a correctly identified setting, our CausalFM prior remains robust w.r.t. specification choices.
* **New computational cost analysis.** We now report the **computational cost**  (**new Tables 5 & 6** and **new Sec. 6.4**) and show that CausalFM is competitive.


We will incorporate all changes into the camera-ready version of our paper. Given these improvements, we are confident that our paper will be a valuable contribution to the causal machine learning literature and a good fit for ICLR 2026\.

---

### Meta-Review · Area_Chair_uS3t · 2025-12-24

**Summary:**

The paper proposes  CausalFM a foundation-model framework for causal inference built on prior-data fitted networks (PFNs). CausalFM pretrains a transformer entirely on synthetic data generated from structural causal model (SCM)–based Bayesian priors, enabling it to perform Bayesian causal inference via in-context learning without retraining on new datasets. The paper shows that for consistent Bayesian causal estimation, the prior must assign zero probability to SCMs that violate identifiability of the target causal query. This is operationalized by constructing SCM priors aligned with known causal structures and identifiability assumptions, allowing a single pretrained model to handle back-door, front-door, and instrumental-variable settings. Relative to recently published work, it generalizes CausalPFN to the IV and front door regime. The rebuttal and discussion highlight why identifiability is crucial relative to prior work that constructs similar estimators without taking this into account (e.g. doPFN). I think it presents a solid contribution to the literature building on this space.

**Reviewer Concerns:**

The paper currently does not provide systematic robustness analyses or diagnostics for detecting when the assumed prior or causal graph is wrong, leaving open questions about graceful degradation and practical reliability. A second weakness concerns scalability (in terms of high dimensional covariates), efficiency, and transparency nor interpretability analyses to explain how the model implicitly selects an adjustment strategy. Finally, the empirical validation is largely synthetic or semi-synthetic; the absence of purely real-world case studies would improve the work.

**Reviewer Scores:**

I felt the author response was thorough. They did provide a comprehensive summary of the concerns raised. I think atleast one of the two reviewers who listed the paper as borderline would have raised their review which informs my current score.

---

### Decision · Program_Chairs · 2026-01-26

Accept (Poster)